# Exponential Graph is Provably Efficient for Decentralized Deep Training

**Bicheng Ying**[1,3]*, **Kun Yuan**[2]*, **Yiming Chen**[2]*, **Hanbin Hu**[4], **Pan Pan**[2], **Wotao Yin**[2]
[1] University of California, Los Angeles [2] DAMO Academy, Alibaba Group
[3] Google Inc. [4] University of California, Santa Barbara
ybc@ucla.edu, {kun.yuan, charles.cym}@alibaba-inc.com,
hanbinhu@ucsb.edu, {panpan.pp, wotao.yin}@alibaba-inc.com

## Abstract

Decentralized SGD is an emerging training method for deep learning known for its much less (thus faster) communication per iteration, which relaxes the averaging step in parallel SGD to inexact averaging. The less exact the averaging is, however, the more the total iterations the training needs to take. Therefore, the key to making decentralized SGD efficient is to realize nearly-exact averaging using little communication. This requires a skillful choice of communication topology, which is an under-studied topic in decentralized optimization.

In this paper, we study so-called exponential graphs where every node is connected to $O(\log(n))$ neighbors and $n$ is the total number of nodes. This work proves such graphs can lead to both fast communication and effective averaging simultaneously. We also discover that a sequence of $\log(n)$ one-peer exponential graphs, in which each node communicates to one single neighbor per iteration, can together achieve exact averaging. This favorable property enables one-peer exponential graph to average as effective as its static counterpart but communicates more efficiently. We apply these exponential graphs in decentralized (momentum) SGD to obtain the state-of-the-art balance between per-iteration communication and iteration complexity among all commonly-used topologies. Experimental results on a variety of tasks and models demonstrate that decentralized (momentum) SGD over exponential graphs promises both fast and high-quality training. Our code is implemented through BlueFog and available at https://github.com/Bluefog-Lib/NeurIPS2021-Exponential-Graph.

## 1 Introduction

Efficient distributed training methods across multiple computing nodes are critical for large-scale modern deep learning tasks. Parallel stochastic gradient descent (SGD) is a widely-used approach, which, at each iteration, computes a globally averaged gradient either using Parameter-Server [28] or All-Reduce [47]. Such global coordination across all nodes in parallel SGD results in either significant bandwidth cost or high latency, which can notably hamper the training scalability.

Decentralized SGD [45, 11, 30, 3] based on *partial averaging* has been one of the promising alternatives to parallel SGD in distributed deep training. Partial averaging, as opposed to the global averaging exploited in parallel SGD, only requires each node to compute the locally averaged model within its neighborhood. Decentralized SGD does not involve any global operations, so it has much lower communication overhead per iteration. The fewer neighbors each node needs to communicate, the more efficient the **per-iteration communication** is in decentralized SGD.

---

*Equal Contribution. Corresponding Author: Wotao Yin

Table 1: Comparison between decentralized (momentum) SGD over (some) various commonly-used topologes. The table assumes homogeneous data distributions across all nodes (which is practical for deep training within a data-center). The comparison for data-heterogeneous scenarios, and with more other topologies, is listed in Appendix C. The smaller the transient iteration complexity is, the faster decentralized algorithms will converge.

| Topology | Ring | Grid | Rand-Graph | Rand-Match | Static Exp | One-peer Exp |
|---|---|---|---|---|---|---|
| **Per-iter Comm.** | $\Omega(2)$ | $\Omega(4)$ | $\Omega(\frac{n}{2})$ | $\Omega(1)$ | $\Omega(\log_2(n))$ | $\Omega(1)$ |
| **Trans. Iters.** | $\Omega(n^7)$ | $\Omega(n^5)$ | $\Omega(n^3)$ | — | $\Omega(n^3 \log_2^2(n))$ | $\Omega(n^3 \log_2^2(n))$ |

The reduced communication in decentralized SGD comes with a cost: slower convergence. While it can asymptotically achieve the same convergence linear speedup as parallel SGD [30, 3, 25, 64], i.e., the training speed increases proportionally to the number of computing nodes (see the definition in Sec. 2), decentralized SGD requires more iterations to reach that stage due to the ineffectiveness to aggregate information using partial averaging. We refer those iterations before decentralized SGD reaches its linear speedup stage as **transient iterations** (see the definition in Sec. 2), which is an important metric to measure the influence of partial-averaging [48, 65] on convergence rate of decentralized SGD. The less effective the partial averaging is, the more transient iterations decentralized SGD needs to take. Fig. 1 illustrates the transient iterations of decentralized SGD for the logistic regression problem. It is observed that decentralized SGD can asymptotically converge as fast as parallel SGD, but it requires more iterations (i.e., transient iterations) to reach that stage.

Per-iteration communication and transient iterations in decentralized SGD are determined by the network topology (we also use graph interchangeably with topology). The maximum degree of the graph decides the communication cost while the connectivity influences the transient iteration complexity. Generally speaking, a sparsely-connected topology communicates cheaply but endows decentralized SGD with more transient iterations due to the less effective information aggregation. A skillful choice of network topology, which is critical to achieve balance between per-iteration communication and transient iteration complexity, is under-studied in literature.

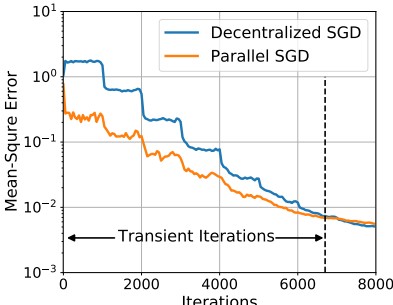

Figure 1: Illustration of transient iters. Experimental setting is in Appendix D.5.

This work studies **exponential graphs** which are empirically successful [3, 61, 27, 14, 67] but less theoretically understood in deep training. Exponential graphs have two variants. In a static exponential graph, each node communicates to $\lceil \log_2(n) \rceil$ neighbors (see Sec. 3 and Fig. 2). In one-peer exponential graph, however, each node cycles through all its neighbors, communicating, only, to a single neighbor per iteration (see Sec. 4 and Fig. 2). This paper will first clarify the connectivity and averaging effectiveness of these exponential graphs, and then apply them to decentralized momentum SGD to obtain the state-of-the-art balance between per-iteration communication and transient iteration complexity among all commonly-used topologies. Our main results (as well as our contributions) are:

- We prove that the spectral gap, which is used to measure the connectivity of the graph (see the definition in Sec. 2), of the static exponential graph is upper bounded by $O(1/\log_2(n))$. Before us, many literatures (e.g. [27]) claimed its upper bound to be $O(1)$ incorrectly.
- Since one-peer exponential graphs are time-varying, it is difficult to derive their spectral gaps. However, we establish that any $\log_2(n)$ consecutive sequence of one-peer exponential graphs can together achieve *exact averaging* when $n$ is a power of $2$.
- With the above results, we establish that one-peer exponential graph, though much sparser than its static counterpart, surprisingly endows decentralized momentum SGD with the **same** convergence rate as static exponential graph in terms of the best-known bounds.
- We derive that exponential graphs achieve $\tilde{\Omega}(1)^2$ per-iteration communication and $\tilde{\Omega}(n^3)$ transient iterations, both of which are nearly the best among other known topologies, see Table 1. The one-peer exponential graph is particularly recommended for decentralized deep training.
- We conduct extensive industry-level experiments across different tasks and models with various decentralized methods, graphs, and network size to validate our theoretical results.

---

[2]Notation $\tilde{\Omega}(\cdot)$ hides all logarithm factors.

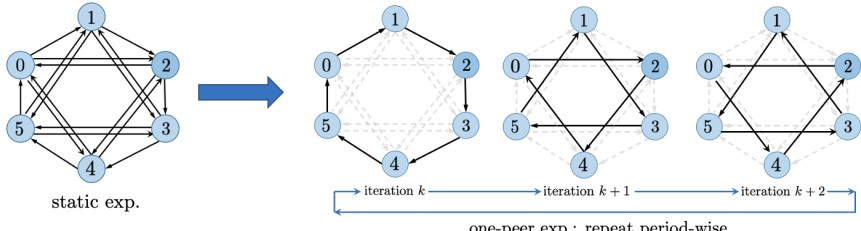

Figure 2: Illustration of the static and one-peer exponential graph.

## 2 Revisit Decentralized Momentum SGD and Related Works

This section reviews basic concepts and existing results on decentralized momentum SGD.

**Problem.** Suppose $n$ computing nodes cooperate to solve the distributed optimization problem:

$$\min_{x \in \mathbb{R}^d} f(x) = \frac{1}{n} \sum_{i=1}^{n} f_i(x) \quad \text{where} \quad f_i(x) := \mathbb{E}_{\xi_i \sim D_i} F(x; \xi_i). \tag{1}$$

Function $f_i(x)$ is local to node $i$, and random variable $\xi_i$ denotes the local data that follows distribution $D_i$. We do not assume each distribution $D_i$ is the same across all nodes.

**Network topology and weights.** Decentralized methods are based on partial averaging within neighborhood that is defined by the network topology (see the figure 2 as an example of six nodes). We assume all computing nodes are connected by a (directed or undirected) network topology. We define $w_{ij}$, the weight to scale information flowing from node $j$ to node $i$, as follows:

$$w_{ij} \begin{cases} > 0 & \text{if node } j \text{ is connected to } i, \text{ or } i = j; \\ = 0 & \text{otherwise.} \end{cases} \tag{2}$$

$\mathcal{N}_i := \{j | w_{ij} > 0\}$ is defined as the set of neighbors of node $i$ which also includes node $i$ itself and the *weight matrix* $W := [w_{ij}]_{i,j=1}^{n} \in \mathbb{R}^{n \times n}$ are denoted as a matrix that stacks the weights of all nodes. This matrix $W$ characterizes the sparsity and connectivity of the underlying network topology.

**Decentralized momentum SGD (DmSGD).** There are many variants of decentralized momentum SGD [3, 20, 32, 67]. This paper will focus on the one proposed by [64] (listed in Algorithm 1), which imposes an additional partial-averaging over the momentum to achieve further speed up. The topology is allowed to change with iterations. When $W^k \equiv W$, topology and weight matrix will remain static.

---

**Algorithm 1** DmSGD

> **Initialize** $\gamma, x_i^{(0)}$; let $m_i^{(0)} = 0, \beta \in (0,1)$
> **For** $k = 0, 1, 2, ..., T-1$, every node $i$ **do**
>> Sample weight matrix $W^{(k)}$;
>> Update gradient $g_i^{(k)} = \nabla F(x_i^{(k)}; \xi_i^{(k)})$;
>> $m_i^{(k+1)} = \sum_{j \in \mathcal{N}_i} w_{ij}^{(k)} (\beta m_j^{(k)} + g_j^{(k)})$;
>> $x_i^{(k+1)} = \sum_{j \in \mathcal{N}_i} w_{ij}^{(k)} (x_j^{(k)} - \gamma m_j^{(k)})$;

---

**Assumptions.** We introduce several standard assumptions to facilitate future analysis:

**A.1** [SMOOTHNESS] Each $f_i(x)$ is $L$-smooth, i.e., $\|\nabla f_i(x) - \nabla f_i(y)\| \le L\|x-y\|$ for any $x, y \in \mathbb{R}^d$.

**A.2** [GRADIENT NOISE] The random sample $\xi_i^{(k)}$ is independent of each other for any $k$ and $i$. We also assume $\mathbb{E}[\nabla F(x; \xi_i)] = \nabla f_i(x)$ and $\mathbb{E}\|\nabla F(x; \xi_i) - \nabla f_i(x)\|^2 \le \sigma^2$.

**A.3** [DATA HETEROGENEITY] It holds that $\frac{1}{n} \sum_{i=1}^{n} \|\nabla f_i(x) - \nabla f(x)\|^2 \le b^2$ for any $x \in \mathbb{R}^d$.

**A.4** [WEIGHT MATRIX] The weight matrix $W^{(k)}$ is doubly-stochastic, i.e. $W^{(k)} \mathbb{1} = \mathbb{1}$ and $\mathbb{1}^T W^{(k)} = \mathbb{1}^T$. If $W^{(k)} \equiv W$, we assume $\rho(W) := \max_{\lambda_i(W) \ne 1} \{|\lambda_i(W)|\} \in (0, 1)$, where $\lambda_i(W)$ is the $i$−th eigenvalue of the matrix $W$.[3]

The quantity $1 - \rho$, which is also referred to as the spectral gap of the weight matrix $W$, measures how well the topology is connected [53]. In the large and sparse topology which is most valuable to deep training, it typically holds that $1 - \rho \to 0$.

---

[3]When there is no ambiguity, we simply use $\rho$ instead of $\rho(W)$. Throughout the paper we do NOT use $\rho$ as spectral radius. Instead, it is the second largest eigenvalue in magnitude. Note we cannot directly sort the eigenvalues since $W$ is not necessarily symmetric and the eigenvalue can be a complex number.

**Communication overhead.** According to [5], global averaging across $n$ nodes either incurs $\Omega(n)$ bandwidth cost via Parameter-Server, or $\Omega(n)$ latency via Ring-Allreduce. In either way, it takes $\Omega(n)$ per-iteration communication time, which is proportional to the network size $n$. As to decentralized methods, we will similarly assume the per-iteration communication time to be $\Omega(\text{maximum degree})$.

**Convergence.** Under Assumptions A.1–A.4, DmSGD with static topology will converge at [64, 25]:

$$\frac{1}{T}\sum_{k=1}^{T}\mathbb{E}\,\|\nabla f(\bar{\mathbf{x}}^{(k)})\|^2 = O\left(\frac{\sigma^2}{\sqrt{nT}} + \frac{n\sigma^2}{T(1-\rho)} + \frac{nb^2}{T(1-\rho)^2}\right) \tag{3}$$

in which $\bar{x}^{(k)} = \frac{1}{n}\sum_{i=1}^{n}x_i^{(k)}$. It is worth noting that no analysis in literature, to our knowledge, exists for DmSGD over time-varying topologies with non-convex costs.

**Linear speedup.** When $T$ is sufficiently large, the first term $1/\sqrt{nT}$ dominates (3). This also applies to parallel SGD. Decentralized and parall SGDs all require $T = \Omega(1/(n\epsilon^2))$ iterations to reach a desired accuracy $\epsilon$, which is inversely proportional to $n$. Therefore, an algorithm is in its linear-speedup stage at $T$th iteration if, for this $T$, the term involving $nT$ is dominating the rate.

**Transient iterations**. Transient iterations are referred to those iterations before an algorithm reaches linear-speedup stage, that is when $T$ is relatively small so non-$nT$ terms still dominate the rate (see illustration in Appendix C). To reach linear speedup, $T$ has to satisfy (derivation in Appendix C)

$$\text{Homogeneous data:}\quad T = \Omega\left(\frac{n^3}{(1-\rho)^2}\right) \qquad \text{Heterogeneous data:}\quad T = \Omega\left(\frac{n^3}{(1-\rho)^4}\right)$$

$$\tag{4}$$

which corresponds to the transient iteration complexity in the homo/hetero-geneous data scenarios.

## 2.1 Related Works

**Decentralized deep training.** Decentralized optimization originates from the control and signal processing community. The first decentralized algorithms on general optimization problems include decentralized gradient descent [45], diffusion [11, 51] and dual averaging [18]. In the deep learning regime, decentralize SGD, which was established in [30] to achieve the same linear speedup as parallel SGD in convergence rate, has attracted a lot of attentions. Many efforts have been made to extend the algorithm to directed topologies [3, 42], time-varying topologies [25, 42], asynchronous settings [31], and data-heterogeneous scenarios [57, 62, 32, 67]. Techniques such as quantization/compression [2, 8, 26, 24, 58, 36], periodic updates [55, 25, 64], and lazy communication [37, 38, 13] were also integrated into decentralized SGD to further reduce communiation overheads.

**Topology influence.** The influence of network topology on decentralized SGD was extensively studied in [25, 51, 66, 45, 42, 27]. All these works indicate that a well-connected topology will significantly accelerate decentralized SGD. Two directions have been explored to relieve the influence of network topology. One line of research proposes new algorithms that are less sensitive to topologies. For example, [66, 23, 65, 57, 1] removed data heterogeneity with bias-correction techniques in [68, 29, 62, 40, 69], and [14, 61, 7, 27] utilized periodic global averaging or multiple partial averaging steps. All these methods have improved topology dependence. The other line is to investigate topologies that enable communication-efficient decentralized optimization. [43, 15] examined various topologies (such as ring, grid, torus, expander, etc.) on averaging effectiveness, which, however, are either communication-costly or averaging-ineffective compared to exponential graphs studied in this paper. [41, 6, 9, 10] studied random graphs (such as Erdos-Renyi random graph and random geometric graph) in which each edge is activated randomly. The randomness of the edge activation can cause a highly unbalanced degrees of each node in the graph, which may significantly affect the efficiency in per-iteration communication.

**Algorithms with time-varying topologies.** Many previous works have studied decentralized algorithms with time-varying topologies. [42] and [44] examined the convergence of decentralized (deterministic) gradient descent and gradient tracking under convex scenarios. [17, 52] investigated gradient tracking under non-convex scenarios, but it did not clarify the influence of the time-varying graphs on convergence rate. In the stochastic scenario, [25] illustrates how decentralized SGD is influenced by time-varying topologies in the non-convex scenario. However, its analysis cannot be directly extended to the decentralized momentum SGD studied in this work.

Another related work is the **Matcha** method [60] based on disjoint matching decomposition sampling. While similar to Matcha, decentralized SGD over one-peer exponential graphs has several fundamental differences. First, one-peer exponential graph is directed while Matcha only supports *undirected* and *symmetric* matching decomposition. Second, the favorable periodic exact-average property of one-peer exponential graphs only holds when sampled *cyclicly*. However, Matcha only supports *independent* and *random* matching samples in analysis. For these reasons, Matcha cannot cover one-peer exponential graphs (especially when momentum is utilized in decentralized SGD).

**Note.** This paper considers deep training within **high-performance data-center clusters**, in which all GPUs are connected with high-bandwidth channels and the network topology can be fully controlled. It is **not** for the wireless network setting in which the topology cannot be changed freely.

## 3  Spectral Gap of Static Exponential Graph

As discussed above, the graph maximum degree decides the per-iteration communication cost while the spectral gap determines the transient iteration complexity (see (4)). It is critical to seek topologies that are both sparse and with large spectral gap $1 - \rho$ simultaneously. In this section, we will establish that the static exponential graph, which was first introduced in [3, 30], is one of such topologies.

In a static exponential graph, each node is assigned an index from 0 to $n - 1$ and will communicate to neighbors that are $2^0, 2^1, \cdots, 2^{\lfloor \log_2(n-1) \rfloor}$ hops away. The left plot in Fig. 2 illustrates a directed 6-node exponential network topology. With maximum degree $\lceil \log_2(n) \rceil$ neighbors, partial averaging over the static exponential graph will take $\Omega(\log_2(n))$ communication time per iteration. However, it remains unclear what the spectral gap is for this topology.

**Weight matrix associated with static exponential graph** is defined as follows:

$$
w_{ij}^{\mathrm{exp}} = \begin{cases} \frac{1}{\lceil \log_2(n) \rceil + 1} & \text{if } \log_2(\mathrm{mod}(j - i, n)) \text{ is an integer or } i = j \\ 0 & \text{otherwise.} \end{cases} \tag{5}
$$

An example weight matrix associated with the static exponential graph in Fig. 2 is in Appendix A.1. The following proposition evaluates the spectral gap $1 - \rho$ for weight matrix in (5).

**Proposition 1** (SPECTRAL GAP OF STATIC EXPO) *The spectral gap of matrix* (5)*, which can also be interpreted as the second largest magnitude of eigenvalues, satisfies (Proof is in Appendix A.2)*

$$
1 - \rho(W^{\mathrm{exp}}) \begin{cases} = \dfrac{2}{1 + \lceil \log_2(n) \rceil}, & \text{when } n \text{ is even number} \\[3mm] < \dfrac{2}{1 + \lceil \log_2(n) \rceil}, & \text{when } n \text{ is odd number} \end{cases} \tag{6}
$$

*In addition, we have* $\|W^{\mathrm{exp}} - \frac{1}{n} \mathbb{1}\mathbb{1}^T\|_2 = \rho(W^{\mathrm{exp}})$.

**Remark 1** *For a general non-symmetric matrix $W$, it typically holds that $\|W - \frac{1}{n}\mathbb{1}\mathbb{1}^T\|_2 \neq \rho(W)$. Proposition 1 establishes $\|W^{\mathrm{exp}} - \frac{1}{n}\mathbb{1}\mathbb{1}^T\|_2 = \rho(W^{\mathrm{exp}})$ for exponential graph.*

**Remark 2** *The hypercube graph is established in [59, Chapter 16] to have the spectral gap as $1 - \rho(W^{\mathrm{HyperCube}}) = 2/(1 + \log_2(n))$. While such spectral gap is on the same order as the exponential graph, there are two fundamental differences between these two graphs: (a) the hypercube graph has to be undirected and the corresponding $W$ is symmetric; (b) the number of vertices of hypercube must be a power of 2, i.e., $n = 2^\tau$ for some positive integer $\tau$. In comparision, the exponential graph is more flexible in the size of the graph structure.*

**Remark 3** *Proposition 1 clarifies the spectral gap of the static exponential graph. Many literatures before this work (such as [27]) claimed the spectral gap to be $O(1)$, which is not accurate.*

The theoretical analysis of Proposition 1 is non-trivial. To evaluate the spectral gap, for any network size $n$, we have to derive the analytical expression for each eigenvalue using Fourier transform and calculate the magnitudes. The most tricky part is to assert which eigenvalue expression attains the *second* largest value.

We now numerically validate the established spectral gap. In Fig. 3, we plotted the spectral gap of the static exponential graph with $n$ ranging from 4 to 290. It is observed that the derived gap $\rho = 1 - 2/(1 + \lceil \log_2(n) \rceil)$ is very tight (see the black dashed line). In fact, it exactly matches the numerical spectral gap when $n$ is even. Moreover, it is also observed the spectral gap of static exponential graph is much smaller than that of ring or grid.

Finally, we compare the spectral gap and maximum degree of the static exponential graph with all other common graphs in Appendix A.3. It is observed that static exponential graph, while with a sightly larger maximum degree, has a significantly smaller spectral gap than ring and grid.

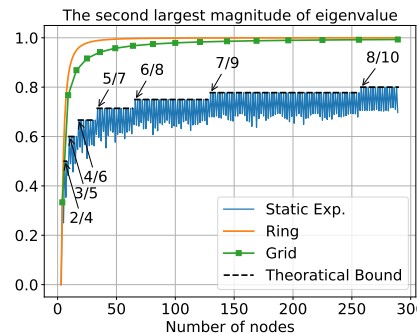

Figure 3: Spectral gap of some topologies.

## 4  One-Peer Exponential Graph Achieves Periodic Exact-Averaging

Static exponential graph incurs $\Omega(\log_2(n))$ communication overhead per iteration. To overcome this issue, [3] proposes to decompose the static exponential graph into a sequence of one-peer graphs, in which each node cycles through all its neighbors, communicating, only, to a single neighbor per iteration, see the right plot in Fig. 2. Apparently, each one-peer realization incurs $\Omega(1)$ communication cost, which matches with ring or grid. Since each realization is sparser than the static graph, one may expect DmSGD with one-peer exponential graphs are less effective in aggregating information. In the following, we will establish an interesting result: one-peer is very effective in averaging.

**Time-varying weight matrix.** We let $\tau = \lceil \log_2(n) \rceil$. The weight matrix at iteration $k$ is

$$w_{ij}^{(k)} = \begin{cases} \frac{1}{2} & \text{if } \log_2(\text{mod}(j - i, n)) = \text{mod}(k, \tau) \\ \frac{1}{2} & \text{if } i = j \\ 0 & \text{otherwise.} \end{cases} \tag{7}$$

The weight matrix for each realization of the one-peer exponential graphs in Fig. 2 is in Appendix B.1. Since each node communicates to one single neighbor per iteration, the resulting weight matrix is very sparse, with only one non-zero element in the non-diagonal positions per row and column.

**Periodic exact-averaging.** The periodic exact-averaging property, which was observed by [3] without theoretical justifications, is fundamental to clarify the averaging effectiveness of one-peer exponential graphs. The following lemma proves that the property holds when $n$ is a power of 2.

**Lemma 1** (PERIODIC EXACT AVERAGING) *Suppose $\tau = \log_2(n)$ is a positive integer. If $W^{(k)}$ is the weight matrix generated by* (7) *over the one-peer exponential graphs, it then holds that each $W^{(k)}$ is doubly-stochastic, i.e. $W^{(k)} \mathbb{1} = \mathbb{1}$ and $\mathbb{1}^T W^{(k)} = \mathbb{1}^T$. Furthermore, it holds that*

$$W^{(k+\ell)} W^{(k+\ell-1)} \cdots W^{(k+1)} W^{(k)} = \frac{1}{n} \mathbb{1} \mathbb{1}^T \tag{8}$$

*for any integer $k \geq 0$ and $\ell \geq \tau$. And equivalently, the consensus residue form holds that*

$$\left( W^{(k+\ell)} - \frac{1}{n} \mathbb{1} \mathbb{1}^T \right) \left( W^{(k+\ell-1)} - \frac{1}{n} \mathbb{1} \mathbb{1}^T \right) \cdots \left( W^{(k)} - \frac{1}{n} \mathbb{1} \mathbb{1}^T \right) = 0 \tag{9}$$

*(Proof is in Appendix B.2).*

**Remark 4** *The assumption that $\log_2(n)$ is a positive integer seems necessary. We numerically tested various one-peer exponential graphs with non-integer $\log_2(n)$. None of them is endowed with the periodic exact-average property.*

**Remark 5** *When $\log_2(n)$ is a positive integer and each realization of the one-peer exponential graph is sampled **without replacement**, it is easy to verify that the periodic exact-averaging property still holds. However, if each realization is sampled with replacement, the periodic exact-averaging property generally does not hold unless all realizations are occasionally sampled without repeating.*

**Remark 6** *It is worth noting that an one-peer variant of the hypercube graph is established to achieve exact averaging with $\tau = \log_n(n)$ steps [54]. Such one-peer hypercube is undirected and symmetric, which is different from the one-peer exponential graph which is directed and asymmetric.*

We now numerically validate Lemma 1. To this end, we initialize a vector $x \in \mathbb{R}^d$ arbitrarily, and examine how $\|(\Pi_{\ell=0}^{k} W^{(\ell)} - \frac{1}{n}\mathbb{1}\mathbb{1}^T)x\|$ decreases with iteration $k$. The weight matrix $W^{(k)}$ is either static or samples from one-peer exponential graph or bipartite random match graph. In Fig. 4, it is observed that one-peer exponential graphs can achieve exact average after $\log_2(n)$ steps, which coincides with the results in Lemma 1. In contrast, the static exponential and bipartite random match graphs can only achieve the global average **asymptotically**. The justification for Remarks 4 and 5 is in Appendix B.3.

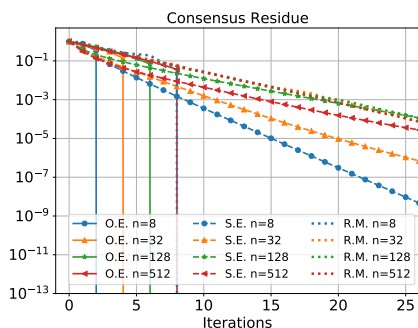

Figure 4: Illustration of how consensus residues decay with iterations for various graphs. O.E. and S.E. denote one-peer and static exponential graphs, and R.M. denotes bipartite random match graph.

## 5 DmSGD with Exponential Graphs

With the derived property in Sec. 3 and 4, this section will examine the convergence of DmSGD with static and one-peer exponential graphs.

**DmSGD with static exponential graph.** Based on Proposition 1, we can achieve the convergence rate and transient iterations, by following analysis in [64], of DmSGD with static exponential graph.

**Corollary 1** *Under Assumptions A.1–A.4, if* $\gamma = \frac{\sqrt{n(1-\beta)^3}}{\sqrt{T}}$, *DmSGD (Algorithm 1) will converge at*

$$\frac{1}{T}\sum_{k=1}^{T}\mathbb{E}\,\|\nabla f(\bar{\mathbf{x}}^{(k)})\|^2 = O\left(\frac{\sigma^2}{\sqrt{(1-\beta)nT}} + \frac{n\log_2(n)(1-\beta)\sigma^2}{T} + \frac{n(1-\beta)b^2\log_2^2(n)}{T}\right) \quad (10)$$

*Furthermore, the transient iteration complexity of DmSGD over static exponential graph is* $O(n^3\log_2^2(n))$ *for data-homogeneous scenario and* $O(n^3\log_2^4(n))$ *for data-heterogeneous scenario.*

**DmSGD with one-peer exponential graph.** With each realization being sparser than its static counterpart, one-peer exponential graph is believed to converge slower. However, the periodic exact-averaging property can help DmSGD achieve the same convergence rate as its static counterpart. Note that DmSGD with one-peer exponential graph is an **one-loop** algorithm, see Algorithm 1. The DmSGD updates start immediately after sampling one weight matrix.

**Theorem 1** *We assume* $\tau = \log_2(n)$ *is a positive integer, and the time-varying weight matrix is generated by* (7) *over one-peer exponential graphs. Under Assumptions A.1–A.4 and* $\gamma = \frac{\sqrt{n(1-\beta)^3}}{\sqrt{T}}$, *DmSGD (Algorithm 1) will converge at (Proof is in Appendix D.1-D.3).*

$$\frac{1}{T}\sum_{k=1}^{T}\mathbb{E}\,\|\nabla f(\bar{\mathbf{x}}^{(k)})\|^2 = O\left(\frac{\sigma^2}{\sqrt{(1-\beta)nT}} + \frac{n(1-\beta)\sigma^2\tau}{T} + \frac{n(1-\beta)b^2\tau^2}{T}\right). \quad (11)$$

*Furthermore, the transient iteration complexity of DmSGD over one-peer exponential graph is* $O(n^3\log_2^2(n))$ *for data-homogeneous scenario and* $O(n^3\log_2^4(n))$ *for data-heterogeneous scenario.*

**Remark 7** *Comparing* (11) *with* (10), *and noting that* $\tau = \log_2(n)$, *we conclude that **DmSGD with one-peer graphs converge exactly as fast as with the static counterpart in terms of the established rate bounds**. In addition, both graphs endow DmSGD with the same transient iteration complexity.*

**Remark 8** *We can also achieve the convergence rate for decentralized SGD (i.e., DSGD without momentum acceleration) with one-peer exponential graph by setting* $\beta = 0$. *It is easy to verify that **DSGD with one-peer graphs can also converge as fast as with the static exponential graph**.*

**Remark 9** *The convergence rate and transient iteration complexity of DSGD with general mixing matrices sampling strategy are also studied in* [25]. *However, the results in reference* [25] *does not cover the scenario with momentum acceleration. As we show in the proof details, it is highly non-trivial to handle momentum.*

It is worth noting that the analysis for the above theorem is non-trivial. While it targets on the one-peer exponential graph, the analysis techniques can be extended to the general time-varying topologies. To our best knowledge, it establishes the first result for DSGD with momentum acceleration, over the time-varying topologies, and in the non-convex settings. Existing analysis either focuses on DSGD without momentum [25], or DmSGD with static topologies [64]. In addition, the last two terms in (11), actually, can be further tightened by the spectral gap of one-peer exponential graphs. Since the tightened terms are rather complicated, we leave them to the discussion in Appendix D.4.

**State-of-the-art balance between communication and convergence.** Table 1 (and tables in Appendix D.5) summarize the per-iteration communication time and transient iteration complexity for all commonly-used topologies. When $n$ is sufficiently large, the term $\log_2(n)$ can be ignored. In this scenario, the exponential graphs (including both static and one-peer variants) achieve state-of-the-art $\tilde{\Omega}(1)$ per-iteration communication and $\tilde{\Omega}(n^3)$ transient iterations, in which $\tilde{\Omega}(\cdot)$ hides all logarithm factors. In Appendix D.5, we numerically validate that exponential graphs have smaller transient iteration complexity than ring or grid graph as predicted in Table 1. The comparison between exponential graph with random graphs [41, 6, 9, 10] (such as the Erdos-Renyi graph and geometric random graph) is discussed in Appendix A.3.3.

**One-peer exponential graph is recommended for decentralized deep training**. It is because one-peer exponential graph endows DmSGD with the same convergence rate as its static counterpart, but incurs strictly less communication overhead per iteration.

# 6 Experiments

This section will validate our theoretical results by extensive deep learning experiments. First, we evaluate how DmSGD with exponential graphs perform against other commonly-used graphs with varying network size. Second, we examine whether one-peer exponential graphs achieve the same convergence rate and accuracy as its static counterpart across different tasks, models, and algorithms.

**Metrics.** Training time and validation accuracy are two critical metrics to examine the effectiveness of a distributed training algorithm in deep learning. These two metrics are typically evaluated after the algorithm completes a fixed number of epochs (say, 90 epochs). Training time can reflect the communication efficiency while accuracy, though might not be precise, can roughly measure the convergence rate (or iteration complexity). These two metrics are used in most of our experiments.

## 6.1 Setup

We implement all decentralized algorithms with PyTorch [46] 1.8.0 using NCCL 2.8.3 (CUDA 10.1) as the communication backend. For parallel SGD, we used PyTorch's native Distributed Data Parallel (DDP) module. For the implementation of decentralized methods, we utilize **BlueFog** [63], which is a high-performance decentralized deep training framework, to facilitate the topology organization, weight matrix generation, and efficient partial averaging. We also follow DDP's design to enable computation and communication overlap. Each server contains 8 V100 GPUs in our cluster and is treated as one node. The inter-node network fabrics are 25 Gbps TCP as default, which is a common distributed training platform setting.

## 6.2 Exponential graphs enable efficient and high-quality training

In this subsection we evaluate how DmSGD with exponential graphs perform against other commonly-used topologies in the task of image classification.

**Implementation.** We conduct a series of image classification experiments with the ImageNet-1K [16], which consists of 1,281,167 training images and 50,000 validation images in 1000 classes. We train classification models with different topologies and numbers of nodes to verify our theoretical findings. The training protocol in [21] is used. In details, we train total 90 epochs. The learning rate is warmed up in the first 5 epochs and is decayed by a factor of 10 at 30, 60 and 80-th epoch. The momentum SGD optimizer is used with linear learning rate scaling by default. Experiments are trained in the mixed precision using Pytorch native amp module. We implement DmSGD with all graphs listed in Table 1. The details of each graph is described in Appendix E. For each graph, we test the training time and validation accuracy for DmSGD with GPU numbers ranging from 32 to 256.

**Experiment results.** The comparison between different graphs (with varying size) in top-1 validation accuracy and training time after 90 epochs is listed in Table 2. Major observations are:

Table 2: Comparison of top-1 validation accuracy(%) and training time (hours) with different topologies.

| NODES | 4(4x8 GPUs) | | 8(8x8 GPUs) | | 16(16x8 GPUs) | | 32(32x8 GPUs) | |
| TOPOLOGY | ACC. | TIME | ACC. | TIME | ACC. | TIME | ACC. | TIME |
|---|---|---|---|---|---|---|---|---|
| RING | 76.13 ±0.023 | 11.6 | 76.07 ±0.013 | 6.5 | 76.08 ±0.026 | 3.3 | 75.58 ±0.021 | 1.8 |
| GRID | 76.08 ±0.007 | 11.6 | 76.35 ±0.037 | 6.7 | 75.88 ±0.011 | 3.4 | 75.76 ±0.022 | 2.0 |
| BI-RAND. MATCH. | 75.96 ±0.032 | **11.1** | 76.26 ±0.027 | **5.7** | 76.07 ±0.012 | **2.8** | 75.83 ±0.029 | **1.5** |
| RANDOM GRAPH | 75.97 ±0.028 | 11.5 | 76.01 ±0.033 | 7.1 | 76.18 ±0.008 | 6.7 | 76.24 ±0.018 | 4.7 |
| STATIC EXP. | 76.21 ±0.028 | 11.6 | 76.32 ±0.037 | 6.9 | 76.30 ±0.007 | 4.1 | 76.28 ±0.020 | 2.5 |
| ONE-PEER EXP. | **76.28 ±0.063** | **11.1** | **76.47 ±0.035** | **5.7** | **76.42 ±0.030** | **2.8** | **76.30 ±0.062** | **1.5** |

**[1]** All graphs (except the random graph) endows DmSGD with training time linear speedup. Among them, bipartite random matching and one-peer exponential graphs achieve the best linear speedup due to their efficient per-iteration communication. However, the accuracy of the matching graph cannot match one-peer exponential graph. The random graph fails to achieve linear speedup because of its extremely expensive communication overheads.

**[2]** In the $32 \times 8$ GPUs scenario, the training time to finish all 90 epochs can be sorted as follows: one-peer ≈ Bi-RandMatch < Ring < Grid < static exponential < random graph, which coincides with the per-iteration communication time listed in Table 1.

**[3]** In the $32 \times 8$ GPUs scenario, the training accuracy achieved by each graph after 90 epochs is sorted as follows: random graph ≈ static exponential ≈ one-peer > Bi-RandMatch > Grid > Ring, which coincides with the transient iteration complexity listed in Table 1. Note that the random graph is rather dense (see the detail in Appendix A.3.1) so it has good accuracy but consumes significant wall-clock time in training.

With the second and third observations, we can find exponential graphs (especially the one-peer exponential graph) can enable both fast and high-quality training performance. We also examined the performance of exponential graphs when $n$ is not a power of 2, see Appendix E.2.

### 6.3 One-peer exponential graph v.s. static exponential graph

In this subsection we will focus on the two exponential graphs studied in this paper. In particular, we will validate that one-peer exponential graph endows DmSGD with the same convergence rate as its static counterpart (i.e., the conclusion in Remark 7) across different tasks, models, and algorithms.

**Comparison across models and algorithms.** Now we compare one-peer and static exponential graphs with different neural network architectures and algorithms. The task is image classification and the setting is the same as in Sec. 6.2. We test both graphs for ResNet [22], MobileNetv2 [50] and EfficientNet [56], which are widely-used models in industry. In addition to the DmSGD algorithm (Algorithm 1) studied in this paper, we also examine how exponential graphs perform with other commonly-used decentralized momentum method: the vanilla DmSGD [3] which does not exchange momentum between neighbors, and QG-DmSGD [32] which adds a quasi-global momentum to relieve the influence of data heterogeneity. We do not examine DecentLaM [67] and D² [57] because both methods require *symmetric* weight matrix during the training process which exponential graphs cannot provide. We also list the performance of parallel SGD using global averaging as one baseline.

Table 3 lists the top-1 validation accuracy comparison across all models and algorithms. In all scenarios, it is observed that both graphs can lead to roughly the same accuracy across models and algorithms. The accuracy difference (DIFF) is marginal. We also depict the convergence curves in training loss and accuracy for DmSGD with both graphs in Fig. 5. It shows both

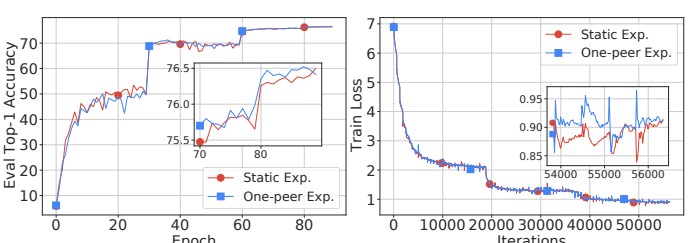

Figure 5: Convergence curves on the ImageNet (ResNet-50) in terms of training loss and validation top-1 accuracy . Network size is $8 \times 8$ GPUs.

curves evolve closely to each other, indicating that one-peer exponential graph enables DmSGD with the same convergence rate as its static counterpart. This is consistent with Theorem 1 and Remark 7. Since one-peer is more communication-efficient than static exponential graph (see Table 2), it is recommended to utilize one-peer exponential graph in decentralized deep training. In addition, we observe that decentralized methods, while utilizing partial-averaging during training process, has no

Table 3: Top-1 validation accuracy and wall-clock time (in hours) comparison with different models and algorithms on ImageNet dataset over static/one-peer exponential graphs (8x8 GPUs).

| MODEL | RESNET-50 | | MOBILENET-V2 | | EFFICIENTNET | |
| TOPOLOGY | STATIC | ONE-PEER | STATIC | ONE-PEER | STATIC | ONE-PEER |
|---|---|---|---|---|---|---|
| PARALLEL SGD | 76.21 (7.0) | - | 70.12 (5.8) | - | 77.63 (9.0) | - |
| VANILLA DMSGD | 76.14 (6.6) | 76.06 (5.5) | 69.98 (5.6) | 69.81 (4.6) | 77.62 (8.4) | 77.48 (6.9) |
| DMSGD | 76.50 (6.9) | 76.52(5.7) | 69.62 (5.7) | 69.98 (4.8) | 77.44 (8.7) | 77.51 (7.1) |
| QG-DMSGD | 76.43 (6.6) | 76.35(5.6) | 69.83 (5.6) | 69.81 (4.6) | 77.60 (8.4) | 77.72 (6.9) |

significant accuracy degradation compared parallel SGD. Decentralized SGD can even be superior sometimes.

**Comparison across different tasks.** We next compare the aforementioned algorithms with one-peer and static exponential graphs in another well-known task: **object detection**. We will test the following widely-used models: Faster-RCNN [49] and RetinaNet [34] on popular PASCAL VOC [19] and COCO [35] datasets. We adopt the MMDetection [12] framework as the building blocks and utilize ResNet-50 with FPN [33] as the backbone network. We choose mean Average Precision (mAP) as the evaluation metric for both datesets. We used 8 GPUs (which are connected by the static or dynamic exponential topology) and set the total batch size as 64 in all detection experiments.

Table 4 compares the performance of decentralized training across different object detection models and datasets. Similar to the above experiment, it is observed that both graphs enable decentralized algorithms with almost the same performance in each scenario. This again illustrates the value of one-peer exponential graph in deep learning tasks - it endows decentralized deep training with both fast training speed and satisfactory accuracy.

Table 4: Comparison of different methods and models on PASCAL VOC and COCO datasets.

| DATASET | PASCAL VOC | | | | COCO | | | |
| MODEL | RETINANET | | FASTER RCNN | | RETINANET | | FASTER RCNN | |
| TOPOLOGY | STATIC | ONE-PEER | STATIC | ONE-PEER | STATIC | ONE-PEER | STATIC | ONE-PEER |
|---|---|---|---|---|---|---|---|---|
| PARALLEL SGD | 79.0 | - | 80.3 | - | 36.2 | - | 37.2 | - |
| VANILLA DMSGD | 79.0 | 79.1 | 80.7 | 80.5 | 36.3 | 36.1 | 37.3 | 37.2 |
| DMSGD | 79.1 | 79.0 | 80.4 | 80.5 | 36.4 | 36.4 | 37.1 | 37.0 |
| QG-DMSGD | 79.2 | 79.1 | 80.8 | 80.4 | 36.3 | 36.2 | 37.2 | 37.1 |

# 7 Conclusion and Future Works

In this paper, we establish the spectral gap of static exponential graph and prove that any $\log_2(n)$ consecutive one-peer exponential graphs can together achieve exact averaging when $n$ is a power of 2. With these results, we reveal that one-peer exponential graphs endow DmSGD with the same convergence rate as their static counterpart. We also establish that exponential graphs achieve nearly minimum per-iteration communication time and transient iteration complexity simultaneously when $n$ is large. All conclusions are thoroughly examined with industrial-standard benchmarks. As the future work, we will investigate symmetric time-varying graphs that can perform as well as one-peer exponential graph. Symmetric graphs are critical for $D^2$ and DecentLaM algorithms.

# Acknowledgements

The authors are grateful to Dr. Sai Praneeth Karimireddy from EPFL for the helpful discussions on the hypercube graph.

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
