# A Static Exponential Graph

## A.1 Weight matrix example

Fig. 6 illustrates the weight matrix $W$ defined in (5) for the 6-node static exponential graph. The four nonzero entries in the first *column* of $W$ correspond to the three outgoing neighbors of node 0 and the node itself; the four nonzero entries on the first *row* corrspond to the three incoming neighbors of node 0 and the node itself.

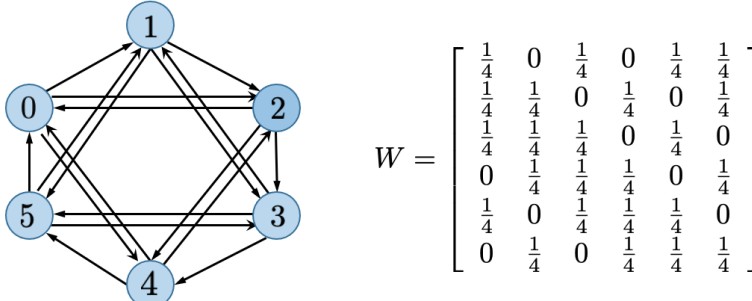

$$
W = \begin{bmatrix}
\frac{1}{4} & 0 & \frac{1}{4} & 0 & \frac{1}{4} & \frac{1}{4} \\
\frac{1}{4} & \frac{1}{4} & 0 & \frac{1}{4} & 0 & \frac{1}{4} \\
\frac{1}{4} & \frac{1}{4} & \frac{1}{4} & 0 & \frac{1}{4} & 0 \\
0 & \frac{1}{4} & \frac{1}{4} & \frac{1}{4} & 0 & \frac{1}{4} \\
\frac{1}{4} & 0 & \frac{1}{4} & \frac{1}{4} & \frac{1}{4} & 0 \\
0 & \frac{1}{4} & 0 & \frac{1}{4} & \frac{1}{4} & \frac{1}{4}
\end{bmatrix}
$$

Figure 6: Illustration of the 6-node static exponential graph and its associated weight matrix.

## A.2 Spectral gap of static exponential graph

Before we present the proof of the spectral gap of static, we first need to review the Discrete Fourier Transform (DFT) and its connection to circulant matrix, which plays the critical role in the proof. We let $\mathrm{Circ}(c_0, c_1, \ldots, c_{n-1})$ denote a circulant matrix, which has the form:

$$
C = \mathrm{Circ}(c_0, c_1, \ldots, c_{n-1}) \triangleq \begin{bmatrix}
c_0 & c_{n-1} & c_{n-2} & \cdots & c_1 \\
c_1 & c_0 & c_{n-1} & & c_2 \\
\vdots & c_1 & c_0 & \ddots & c_{n-2} \\
c_{n-1} & & \ddots & \ddots & c_{n-1} \\
c_n & c_{n-1} & \cdots & c_1 & c_0
\end{bmatrix} \tag{12}
$$

and we call the circulant matrix $C$ is generated by the vector $c = (c_0, c_1, c_2, \ldots, c_{n-1})$. With this notation, the circulant convolution can be equivalently re-written as the matrix-vector multiplication. Suppose we have two vectors $c \in \mathbb{C}^n$ and $v \in \mathbb{C}^n$:

$$
c \otimes v = Cv \tag{13}
$$

where $\otimes$ means the $n$-point circular convolution and $C$ is the circulant matrix generated by vector $c$.

**Lemma 2 (Eigenvalue of circulant matrix)** *The eigenvalues of a circulant matrix* $\mathrm{Circ}(c_0, c_1, \ldots, c_{n-1})$ *are given by*

$$
\lambda_i = c_0 + c_1 \omega^i + c_1 \omega^{2i} + \cdots + c_{n-1} \omega^{(n-1)i}, \quad i = 0, 1, \cdots, n-1 \tag{14}
$$

*where $\omega_i$ is the $i$-th root of unity under $n$-order, i.e., $\omega_i = \exp(2\pi j \frac{i}{n})$. (Note we use $j$ for imaginary number instead of $i$.)*

**Proof**. From the convolution theorem of Discrete Fourier Transform (DFT), we know that, for arbitrary $n$-dimension vector $c$ and $v$, DFT of the $n-$circulant convolution of $c$ and $v$ equals to the element-wise multiplication of DFT of $c$ and DFT of $v$:

$$
\mathfrak{F}(c \otimes v) = \mathfrak{F}(c) \odot \mathfrak{F}(v) \quad \forall c, v \tag{15}
$$

where $\odot$ means the Hadamard product. Introduce the DFT Matrix:

$$\mathcal{F} \triangleq \begin{bmatrix} 1 & 1 & 1 & \cdots & 1 \\ 1 & \omega^1 & \omega^2 & & \omega^{n-1} \\ 1 & \omega^2 & \omega^4 & & \omega^{2(n-1)} \\ \vdots & \vdots & \vdots & \ddots & \vdots \\ 1 & \omega^{n-1} & \cdots & & \omega^{(n-1)^2} \end{bmatrix} \tag{16}$$

It can be verified that

$$\mathrm{Diag}(\mathcal{F}c)\mathcal{F}v = (\mathcal{F}c) \odot (\mathcal{F}v) = \mathcal{F}(c \otimes v) = \mathcal{F}Cv = \frac{1}{n}\mathcal{F}C\mathcal{F}^{\dagger}\mathcal{F}v \tag{17}$$

where $\mathrm{Diag}(x)$ means the diagonal matrix built from vector $x$ and $\mathcal{F}^{\dagger}$ denotes the conjugate transpose of $\mathcal{F}$. In (17), we utilize the two identities that $x \odot y = \mathrm{Diag}(x)y$ and $x \otimes y = Xy$, where $X$ is the circulant matrix generated by $x$. Since (17) holds for any vector $v$, we must have

$$\left(\frac{1}{\sqrt{n}}\mathcal{F}\right)C\left(\frac{1}{\sqrt{n}}\mathcal{F}^{\dagger}\right) = \mathrm{Diag}(\mathcal{F}c) \tag{18}$$

Hence, any circulant matrix $C$ can be diagonalized by DFT matrix $\mathcal{F}$ and the corresponding eigenvalues are the DFT of the generating vector $c$. Expanding the expression for each element in $\mathcal{F}c$ will lead to (14) immediately. ∎.

With this powerful tool, we are ready to prove the spectral gap of exponential graph in Proposition 1. To make the proof easier to follow, we split the proof into two parts. The first part is for special $n = 2^{\tau}$ case. After that, we present the proof for arbitrary number $n$.

**Proof of Proposition 1 (special $n = 2^{\tau}$ case).** First, we note by definition the combination matrix $W^{\mathrm{exp}}$ is a circulant matrix:

$$W^{\mathrm{exp}} = \mathrm{Circ}\left(\frac{1}{\tau+1}, \frac{1}{\tau+1}, \ldots, 0, \frac{1}{\tau+1}, 0, \ldots\right) \tag{19}$$

Resorting to lemma 2, we can immediate conclude that all eigenvalues of exponential graph have the following form:

$$\lambda_i = \frac{1}{\tau+1} + \frac{1}{\tau+1}\omega_i + \frac{1}{\tau+1}\omega_i^2 + \frac{1}{\tau+1}\omega_i^{2^2} + \ldots + \frac{1}{\tau+1}\omega_i^{2^{\tau-1}}, \quad i = 0, 1, ..., N-1 \tag{20}$$

where $\omega_i = \exp\left(2\pi j \frac{i}{N}\right)$ is the $i$-th root of unity under $N$-order. The magnitude of each eigenvalue is:

$$|\lambda_i| = \sqrt{\left(\frac{1}{\tau+1} + \frac{1}{\tau+1}\sum_{n=0}^{\tau-1}\cos(\frac{2\pi i}{N}2^n)\right)^2 + \left(\frac{1}{\tau+1}\sum_{n=0}^{\tau-1}\sin(\frac{2\pi i}{N}2^n)\right)^2} \tag{21}$$

However, it is not obvious which eigenvalue has the second largest magnitude. It is easy to see that

$$\lambda_0 = \frac{1}{\tau+1} + \frac{1}{\tau+1}1 + \ldots + \frac{1}{\tau+1}1 = 1 \tag{22}$$

Recall that the eigenvalues of doubly stochastic matrix $W$ must be equal to or smaller than 1, we know $\lambda_0$ is the largest eigenvalue in magnitude. Next, it is also not hard to check that

$$\begin{aligned} \lambda_{n/2} &= \frac{1}{\tau+1} + \frac{1}{\tau+1}\omega_{n/2} + \ldots + \frac{1}{\tau+1}\omega_{n/2}^{2^{\tau-1}} \\ &= \frac{1}{\tau+1} + \frac{1}{\tau+1}(-1) + \frac{1}{\tau+1}(-1)^2 \ldots + \frac{1}{\tau+1}(-1)^{2^{\tau-1}} \\ &= \frac{\tau-1}{\tau+1} \end{aligned} \tag{23}$$

where $n/2$ must be integer since $n = 2^{\tau}$. As long as we can show that there is no other eigenvalue $\lambda_i$ lying between $\frac{\tau-1}{\tau+1}$ and 1, we can claim that $\frac{2}{\tau+1}$ is the spectral gap of exponential graph.

Consider two cases for the rest $\lambda_i$:

1. If $i$ is an odd number, we know that :

$$\omega_i^{2^{\tau-1}} = \exp(2\pi j \frac{i2^{\tau-1}}{2^\tau}) = \exp(2\pi j \frac{i}{2}) = (-1)^i = -1 \tag{24}$$

Then applying the triangle inequality, we know that

$$|\lambda_i| = \underbrace{\left| \frac{1}{\tau+1}\omega_i + \frac{1}{\tau+1}\omega_i^2 + \frac{1}{\tau+1}\omega_i^{2^2} + \ldots + \frac{1}{\tau+1}\omega_i^{2^{\tau-2}} \right|}_{\tau-1 \text{ terms}} \leq \frac{\tau-1}{\tau+1} \tag{25}$$

Therefore, we conclude that the magnitude of any $|\lambda_i|$, where $i$ is an odd number, must not lie between $\frac{\tau-1}{\tau+1}$ and 1.

2. If $i$ is an even number and $i$ is not zero. We can assume its prime factor decomposition has the following format:

$$i = 2^{t'} p_1^{t_1} p_2^{t_2} \cdots p_\ell^{t_\ell} \tag{26}$$

where $p_\ell$ is some prime number except 2 and $t_\ell$ is the corresponding order. Because we know $i$ is strictly smaller than $2^\tau$ and 2 is smallest prime number, we can claim that $t' \leq \tau - 1$. Since $i$ is some even number larger than 0, we also know $t' > 0$. These two conditions implies that among the index set $\{0, 1, 2, \cdots, \tau - 1\}$, we can always find a number $\tau'$ such that $\tau' + t' = \tau - 1$. We evaluate :

$$\begin{aligned} \omega_i^{2^{\tau'}} &= \exp\left(2\pi j \frac{i2^{\tau'}}{2^\tau}\right) \\ &= \exp\left(2\pi j \frac{2^{\tau'} 2^{t'} p_1^{t_1} p_2^{t_2} \cdots p_\ell^{t_\ell}}{2^\tau}\right) \\ &= \exp\left(2\pi j \frac{p_1^{t_1} p_2^{t_2} \cdots p_\ell^{t_\ell}}{2}\right) \\ &= -1 \end{aligned} \tag{27}$$

Again, using the triangle inequality, we also can conclude that the magnitude of any $|\lambda_i|$, where $i$ is an even number, must not lie between $\frac{\tau-1}{\tau+1}$ and 1.

So combining above two cases, we complete the proof that there is no other eigenvalue having the magnitude that is larger than $(\tau - 1)/(\tau + 1)$ and smaller than 1. ∎

**Proof of Proposition 1 (the general cases).** The first several steps are the same as we did in previous proof. Next, we just need to show that there is no eigenvalue lying between $\frac{\tau-1}{\tau+1}$ and 1.

Among in the index set $\{0, 1, 2, \cdots, \tau - 1\}$, we select two numbers, denoting them as set $S$. Using the triangle inequality, we have

$$\begin{aligned} |\lambda_i| &\leq \frac{1}{\tau+1} \left| \sum_{t=0, t\notin S}^{\tau-1} w_i^{2^t} \right| + \frac{1}{\tau+1} \left| 1 + \sum_{t \in S} w_i^{2^t} \right| \\ &\leq \frac{\tau-2}{\tau+1} + \frac{1}{\tau+1} \left| 1 + \sum_{t \in S} w_i^{2^t} \right|, \quad \forall i \end{aligned} \tag{28}$$

As long as we show that for all feasible $i$ but 0, there always exist a set $S$ such that

$$\left| 1 + \sum_{t \in S} w_i^{2^t} \right| \leq 1 \tag{29}$$

we establish the upper bound for the second largest eigenvalue. The key to solve it is notice that, for $\alpha \in [0.25, 0.75]$, we have

$$|1 + e^{2\pi j\alpha} + e^{2\pi j2\alpha}| = |e^{2\pi j\alpha}(e^{-2\pi j\alpha} + 1 + e^{2\pi j\alpha})| = |1 + \cos(2\pi\alpha)| \leq 1 \tag{30}$$

Next, we discuss case-by-case:

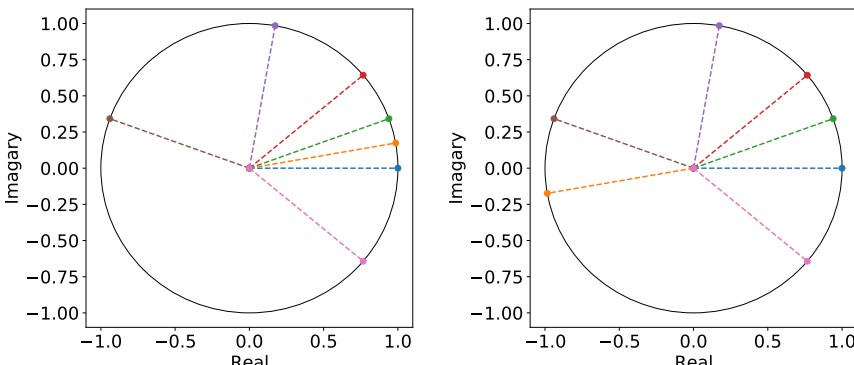

Figure 7: The position of each $\{\omega_i^{2^t}\}_{t=0}^{\tau-1}$ in the complex plane. The left figure shows the case that $i = 1$ and $n = 36$ and the right one shows the case that $i = 19$ and $n = 36$.

- If $\frac{3n}{4} \geq i \geq \frac{n}{4}$, we can simply choose $S = \{0, 1\}$. We know

$$\left| 1 + w_i + w_i^2 \right| = \left| 1 + \exp(2\pi j \frac{i}{n}) + \exp(2\pi j \frac{2i}{n}) \right| \leq 1 \tag{31}$$

- If $\frac{n}{4} > i > 1$. Because $2^\tau \geq n$, there exist a $t$ satisfying $\frac{3n}{4} \geq i2^t \geq \frac{n}{4}$ and $t \leq \tau - 2$. Choosing $S = \{t, t+1\}$ yields the desired inequality as we have in (31): $\left| 1 + \sum_{t \in S} w_i^{2^t} \right| \leq 1$

- If $n - 1 > i > \frac{3n}{4}$. Due the circular symmetry, we know that $w_i = w_{n-i}^*$, where we use $x^*$ as the conjugate of complex number $x$. It implies

$$\begin{aligned} \left| 1 + w_i + w_i^2 \right| &= \left| 1 + w_{n-i}^* + (w_{n-i}^{n-2})^* \right| \\ &= \left| 1 + w_{n-i} + w_{n-i}^{n-2} \right| \end{aligned} \tag{32}$$

  Notice $n - i$ belongs to the range $(1, \frac{n}{4})$, we can immediate conclude that (32) is smaller than 1.

- If $i = 1$. We need to use a different argument to select the index set $S$ since the $t$ satisfying $\frac{3n}{4} \geq i2^t \geq \frac{n}{4}$ may equal $\tau - 1$. However, we still can select $S = \{\tau - 2, \tau - 1\}$:

$$\begin{aligned} \left| 1 + \exp^{2\pi j \frac{2^{\tau-2}}{n}} + \exp^{2\pi j \frac{2^{\tau-1}}{n}} \right| &= \left| \exp^{-2\pi j \frac{2^{\tau-2}}{n}} + 1 + \exp^{2\pi j \frac{2^{\tau-2}}{n}} \right| \\ &= \left| 1 + \cos\left(2\pi \frac{2^{\tau-2}}{n}\right) \right| \end{aligned} \tag{33}$$

  Since $2^\tau \geq n$, we know $\frac{2^{\tau-2}}{n} \geq \frac{1}{4}$. And we also know $\frac{2^{\tau-2}}{n} \leq \frac{1}{2}$, otherwise it indicates that $2^{\tau-1} \geq n$, which contradicts the assumption that $\tau = \lceil \log_2(n) \rceil$. Hence, we can conclude that $\cos\left(2\pi \frac{2^{\tau-2}}{n}\right) \leq 0$.

- If $i = n - 1$. Use the conjugate argument then apply the similar procedure as $i = 1$.

Above 5 cases cover all possible choices of $i$ for eigenvalues(except 0). Hence, we conclude that for arbitrary $n$, the second largest magnitude of eigenvalue of exponential graph is bounded by $\frac{\tau-1}{\tau+1}$.

This bound is attained if $n$ is an even number, $\lambda_{n/2}$ is that desired eigenvalue. Based on the numerical experiment, we know it if $n$ is an odd number, this bound cannot be attained. Unfortunately, we neither have a closed form solution for the spectral gap nor know which eigenvalue will becomes the second largest magnitude of eigenvalue.

Lastly, we need to show that the $\ell_2$ matrix norm $\|W^{\mathrm{exp}} - \frac{1}{n}\mathbb{1}\mathbb{1}^T\|_2^2$ is the same value as spectral gap. To prove that, we resort to the Discrete Fourier Transform again

$$
\begin{aligned}
\|W^{\mathrm{exp}} - \frac{1}{n}\mathbb{1}\mathbb{1}^T\|_2^2 &\overset{(a)}{=} \mathrm{eig}_1\Big((W^{\mathrm{exp}})^T W^{\mathrm{exp}} - \frac{1}{n}\mathbb{1}\mathbb{1}^T\Big)\\
&\overset{(b)}{=} \mathrm{eig}_1\Big(\frac{1}{n^2}\mathcal{F}D^\dagger \mathcal{F}^\dagger \mathcal{F} D \mathcal{F}^\dagger - \frac{1}{n}\mathbb{1}\mathbb{1}^T\Big)\\
&\overset{(c)}{=} \mathrm{eig}_1\Big(\frac{1}{n}\mathcal{F}\big(D^\dagger D - \mathrm{Diag}\{1,0,0,\cdots,0\}\big)\mathcal{F}^\dagger\Big)\\
&= \mathrm{eig}_1\Big(\big(D^\dagger D - \mathrm{Diag}\{1,0,0,\cdots,0\}\big)\Big)\\
&= \rho(W^{\mathrm{exp}})^2
\end{aligned}
\tag{34}
$$

where $\mathrm{eig}_1(\cdot)$ means the largest eigenvalue of matrix, step (a) is because $\|X\|_2^2 = \|X^T X\|_2 = \mathrm{eig}_1(X^T X)$, step (b) applied the eigenvalue decomposition of $W^{\mathrm{exp}}$ through the DFT, step (c) follows the fact that $\mathbb{1}$ is the first column of $\mathcal{F}$. ■

### A.3 Comparing static exponential graph with commonly-used topologies

### A.3.1 Details of each graph and the associated weight matrix

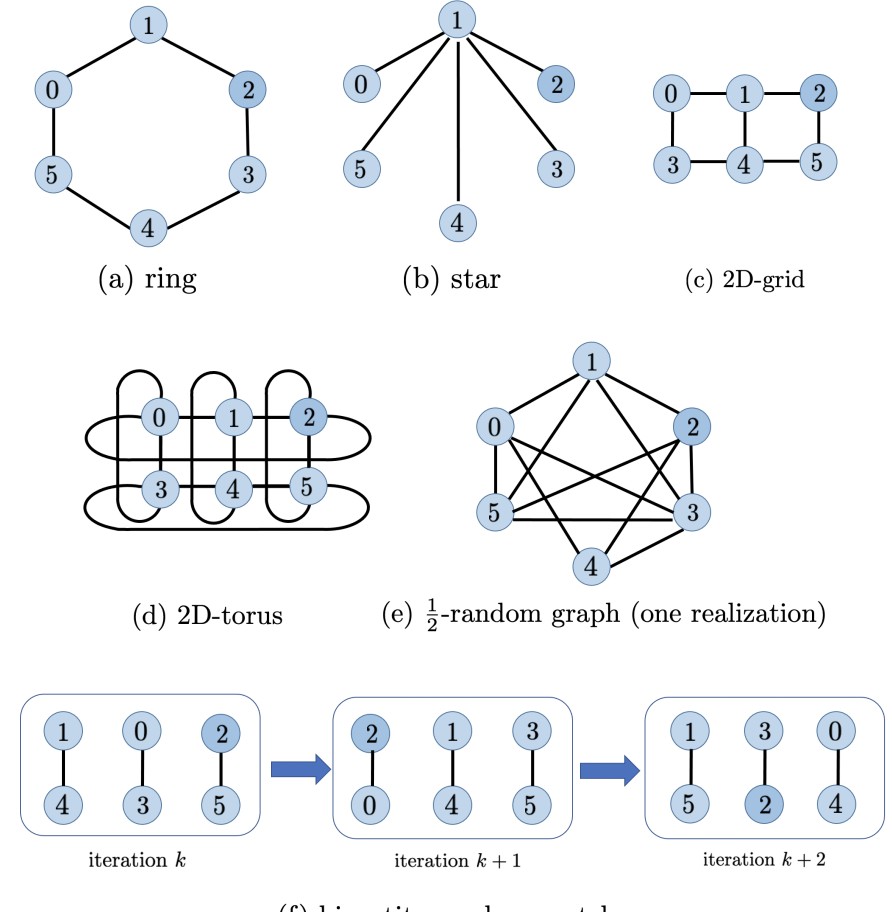

(a) ring      (b) star      (c) 2D-grid

(d) 2D-torus      (e) $\frac{1}{2}$-random graph (one realization)

(f) bipartite random match

Figure 8: The shape of the 6-node topologies discussed in Sec. A.3.1.

- **Ring.** The ring topology is undirected, and is illustrated in Fig. 8(a). Its weight matrix is generated according to the Metropolis rule [43, Eq. (8)], which is symmetric.

- **Star.** The star topology is undirected, and is illustrated in Fig. 8(b). Its weight matrix is generated according to the Metropolis rule, which is symmetric. Note that DmSGD with star graph still conducts partial averaging per iteration. It is different from parallel SGD that utilizes parameter-server (which is also of the star shape) to conduct global averaging.

- **2D-grid.** The 2D-grid topology is undirected, and is illustrated in Fig. 8(c). Its weight matrix is generated according to the Metropolis rule, which is symmetric.

- **2D-torus.** The 2D-torus topology is undirected, and is illustrated in Fig. 8(d). Its weight matrix is generated according to the Metropolis rule, which is symmetric.

- $\frac{1}{2}$-**random graph.** Consider a random $n$-node graph generated with each edge populating independently with probability $p = \frac{1}{2}$. Let $A \in \mathbb{R}^{n \times n}$ be the symmetric adjacency matrix of the graph, with $A_{ij} = 1$ if nodes $i$ and $j$ are connected, and $0$ otherwise. We let $W = A/d_{\max}$ where $d_{\max} = \max_i \sum_j A_{ij}$. By union bound and Bernstein's inequality, one can easily derive $d_{\max}$ concentrates around $(n-1)/2$ with probability 1. It is derived in [43, Proprosition 5] that the $\frac{1}{2}$-random graph has $1 - \rho = O(1)$. A realization of one $\frac{1}{2}$-random graph is depicted in Fig. 8(e). It is observed that the random graph is rather dense.

- **Bipartite random match graph.** We assume $n$ is even. Bipartite random match graph is undirected and time-varying. To generate such graph at iteration $k$, we first randomly permute the index $(1, 2, \ldots, n)$ to $(\sigma^{(k)}(1), \sigma^{(k)}(2), \ldots, \sigma^{(k)}(n))$, where $\sigma^{(k)}(\cdot)$ denotes the permutation function at iteration $k$. Next we let node $\sigma^{(k)}(2j + 1)$ be the neighbor of node $\sigma^{(k)}(2j)$ for each $j = 0, \cdots, n/2 - 1$. It is obvious that, at iteration $k$, each node only exchanges information with one neighbor. The spectral gap of the bipartite random match graph, to our knowledge, is unknown yet in literature. Fig. 8(f) illustrates a sequence of the bipartite random match graphs.

### A.3.2 Comparison with commonly-used topologies

Table 5 summarizes the maximum degree and $1 - \rho$ for commonly-used graphs. The spectral gaps of ring, star, grid and torus are discussed in [43, Proprosition 5].

Table 5: Comparison between commonly-used topologies in maximum degree and $1 - \rho$.

|  | $1 - \rho$ | **Max-degree** |
|---|---|---|
| **ring** | $O(\frac{1}{n^2})$ | 2 |
| **star** | $O(\frac{1}{n^2})$ | $n - 1$ |
| **2D-grid** | $O(\frac{1}{n \log_2(n)})$ | 4 |
| **2D-torus** | $O(\frac{1}{n})$ | 4 |
| $\frac{1}{2}$-**random graph** | $O(1)$ | $\frac{n-1}{2}$ |
| **random match** | N.A. | 1 |
| **static exponential** | $O(\frac{1}{\log_2(n)})$ | $\log_2(n)$ |

### A.3.3 Comparison with random topologies

A random graph is achieved by starting with a set of $n$ nodes and imposing successive edges between them randomly. Random graph is extensively studied in wireless networks. To show the comparison between the exponential graph and various random graphs studied in [41, 6, 9, 10], we first summarize the differences between scenarios in deep learning and in wireless network and control theory:

- **Topology size.** The GPUs utilized in deep learning are typically very expensive. A topology with tens or hundreds of GPUs is already regarded as a large network. This is different from wireless networks which may consist of thousands of (relatively cheap) sensors or mobile agents. The properties that are very likely to hold for large networks with thousands of nodes (e.g. the connectivity of the random graphs in [41, 6, 9, 10] with such a large size) may not valid for network with a small or moderate size.

Table 6: Comparison between exponential graph and the random graphs.

| | E.-R. Random | Geometric Random | Static Exp. | O.P. Exp |
|---|---|---|---|---|
| Per-iter. comm. | $\tilde{\Omega}(1)$ in expectation | $\tilde{\Omega}(1)$ in expectation | $\tilde{\Omega}(1)$ | $\Omega(1)$ |
| Transient iter. | $\tilde{\Omega}(n^3)$ | $\tilde{\Omega}(n^5)$ | $\tilde{\Omega}(n^3)$ | $\tilde{\Omega}(n^3)$ |
| Connectivity | Connected when $n$ is sufficiently large | Connected when $n$ is sufficiently large | Always Connected | Disconnected for some iter.[†] |
| Degree balance | Can be highly unbalanced | Can be highly unbalanced | Balanced | Balanced |

[†]While disconnected for some iteration, it is proved to work for DmSGD.

- **Topology control.** Decentralized deep learning is typically conducted in data-center GPU clusters. In these clusters, GPUs are connected with high-bandwidth channels (such as InfiniBand, the optical fiber, etc.), and they can be organized in any topology shape. However, the network connectivity in the wireless network is highly sensitive to the geographical location of the nodes, and the radius of their wireless signals. The topology cannot be controlled freely in the latter setting.

- **Balanced degree.** Since the topology is in full control for deep learning, the topology design is very important for communication efficiency. In deep learning, we prefer topologies in which all nodes have identical degrees (i.e., the number of neighbors) so that they can finish the communication almost at the same time without waiting for the slowest one. Static and one-peer exponential graphs studied in our paper are such topologies. However, for the random graph in references [41, 6, 9, 10], there always exists the possibility to generate a realization with highly unbalanced degrees, especially when the network size is not large.

References [41, 6, 9, 10] studied various random graphs. In this subsection, we will focus on the Erdos-Renyi graph $G(n,p)$ with $p = (1+c)\log(n)/n$ for some $c > 0$, and the 2-D geometric random graph $G(n,r)$ with $r^2 = (1+c)\log(n)/n$ for some $c > 0$. Both random graphs are widely used in wireless networks. Table 6 lists the comparison between exponential graph and the random graphs. In the table, it is observed that the E.-R. and geometric random graphs are either equivalent to, or worse than, exponential graphs in either per-iteration communication, or the transient iteration complexity. Moreover, note that the per-iteration communication cost for random graphs is calculated in expectation. In practice, the maximum degree in both random graphs must be greater than the expected degree for each node in the table, which will lead to an even slower per-iteration communication cost than exponential graphs. With the results listed in the above table as well as the other comparison described below, we still recommend using exponential graphs in deep learning.

## B  One-peer Exponential Graph

### B.1  Weight matrix example

Fig. 9 illustrates the weight matrix $W$ defined in (7) for the 6-node one-peer exponential graph.

### B.2  Periodic exact averaging of one-peer exponential graph

We present and prove a lemma that is more general than Lemma 1 in the main body. Therefore, its proof also serves the proof of Lemma 1.

**Lemma 3** (EXACT AVERAGING) *Suppose $W^{(k)}$, $k \geq 0$, are the weight matrices defined in* (7) *over the one-peer exponential graph. It holds that each $W^{(k)}$ is doubly-stochastic, i.e., $W^{(k)}\mathbb{1} = \mathbb{1}$ and $\mathbb{1}^T W^{(k)} = \mathbb{1}^T$. Furthermore, if there exists an integer $\tau \geq 0$ such that $n = 2^\tau$, then it holds that*

$$W^{(k_\ell)} \cdots W^{(k_2)} W^{(k_1)} = \frac{1}{n}\mathbb{1}\mathbb{1}^T \tag{35}$$

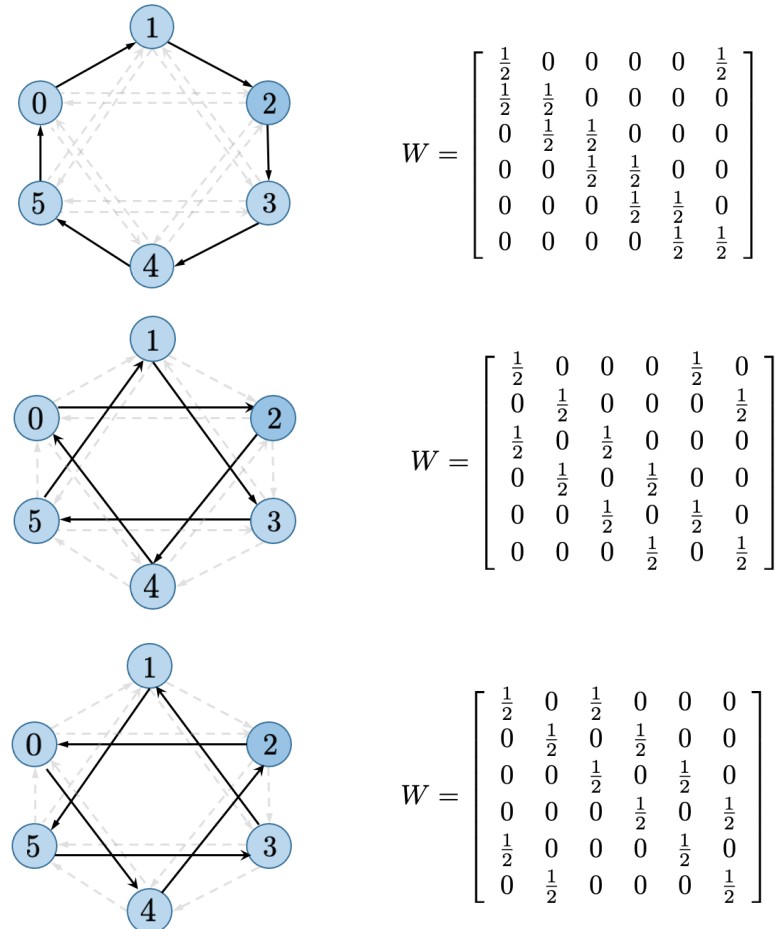

Figure 9: Illustration of the 6-node one-peer exponential graph and its associated weight matrix.

as long as $\{\mathrm{mod}(k_1, \tau), \ldots, \mathrm{mod}(k_\ell, \tau)\} = \{0, \ldots, \tau - 1\}$. *In particular, the weight matrices associated with one-peer exponential graph can help reach an exact consensus average after all $\tau$ different matrices are each applied at least once.*

**Proof.** The double stochasticity of every $W^{(k)}$ follows directly from their definitions. It is left to establish (35).

Since $W^{(k_1)} = W^{(k_2)}$ as long as $\mathrm{mod}(k_1, \tau) = \mathrm{mod}(k_2, \tau)$, we can assume all $k_i \in \{0, \ldots, \tau - 1\}$ without loss of generality.

Since the eigenvectors of all circulant matrices of the same size are the same set of Fourier modes, circulant matrices $W^{(k)}$ for all $k \geq 0$ are simultaneously diagonalizable and their multiplications are commutative, i.e., $W^{(k_1)} W^{(k_2)} = W^{(k_2)} W^{(k_1)}$ for any $k_1, k_2 \geq 0$. This property, together with the fact $\{k_1, \ldots, k_\ell\} = 0, \ldots, \tau - 1$ and the double stochasticity of every $W^{(k)}$, implies it suffices to show $W^{(0)} W^{(1)} \ldots W^{(\tau-1)} = \frac{1}{n} \mathbb{1} \mathbb{1}^T$ or, for the convenience of argument below,

$$(W^{(\tau-1)})^T \ldots (W^{(1)})^T (W^{(0)})^T = \frac{1}{n} \mathbb{1} \mathbb{1}^T. \tag{36}$$

Consider $y = (W^{(\tau-1)})^T \ldots (W^{(1)})^T (W^{(0)})^T x$. (We index the entries of $x, y$ starting from 0, instead of 1, for we use a binary representation below.) Since all nodes are treated equally, it suffices to show $y_0 = \frac{1}{n}(x_0 + \cdots + x_{n-1})$ since, through shifting the node indices, this equality implies $y_i = \frac{1}{n}(x_i + \cdots + x_{n-1} + x_0 + \cdots + x_{i-1}) = \frac{1}{n}(x_0 + \cdots + x_{n-1})$ for $i = 1, 2, \ldots, n - 1$.

In a graph with $n = 2^\tau$ nodes, *index* the nodes by decimal numbers $0, \ldots, n-1$. Obtain their binary-form numbers:

$$
\begin{array}{c}
\text{decimal index} \mid \text{binary index} \\
0 = 0 \ldots 00\mathrm{b} \\
1 = 0 \ldots 01\mathrm{b} \\
2 = 0 \ldots 10\mathrm{b} \\
\vdots \\
n - 2 = 1 \ldots 10\mathrm{b} \\
n - 1 = 1 \ldots 11\mathrm{b}.
\end{array}
$$

There are $\tau$ bits in each binary number above, denoted by $b$; the $j$th bit, $j = 0, \ldots, \tau - 1$, from right to left is denoted by $b_j$. For the example of node 2, $b_0 = 0$, $b_1 = 1$, and then $b_j = 0$ for $j = 2, \ldots, \tau - 1$.

Pick any single $(W^{(k)})^T$ from $k = 0, \ldots, \tau - 1$. The results of applying $x' = (W^{(k)})^T x$ are

$$
y_b = \frac{1}{2}(x_{\mathrm{mod}(b,n)} + x_{\mathrm{mod}(b+2^k,n)}), \quad \forall b = 0, \ldots, n - 1.
$$

In particular, $x' = \frac{1}{2}\sum_{b \in B} x_b$ for $B = \{b : b_j = 0 \ \forall j \neq k\} = \{0, 2^k\}$, that is, all bits of $b$ are 0 except for the $k$th bit, which is either 0 or 1.

Now pick $k_1 \neq k_2 \in \{0, \ldots, \tau - 1\}$, then $x'' = (W^{(k_2)})^T (W^{(k_1)})^T x$ satisfies

$$
\begin{aligned}
x''_b &= \frac{1}{2}(x'_{\mathrm{mod}(b,n)} + x'_{\mathrm{mod}(b+2^{k_2},n)}) \quad \text{where } x' = (W^{(k_1)})^T x \\
&= \frac{1}{4}(x_{\mathrm{mod}(b,n)} + x_{\mathrm{mod}(b+2^{k_1},n)} + x_{\mathrm{mod}(b+2^{k_2},n)} + x_{\mathrm{mod}(b+2^{k_1}+2^{k_2},n)}), \quad \forall b = 0, \ldots, n - 1.
\end{aligned}
$$

In particular, $y_0 = \frac{1}{2}\sum_{b \in B} x_b$ for $B = \{b : b_j = 0 \ \forall j \neq k_1, k_2\}$, that is, all bits of $b$ are 0 except for the $k_1$th and $k_2$th bits, which are either 0 or 1.

Using proof by induction, it is easy to show that $z = (W^{(k_\ell)})^T \ldots (W^{(k_1)})^T x$ for *distinct* $k_\ell, \ldots, k_1 \in \{0, \ldots, \tau - 1\}$ satisfies

$$
y_0 = \frac{1}{2^\ell} \sum_{b \in B} x_b, \quad B = \{b : b_j = 0, \forall j \notin \{k_1, \ldots, k_\ell\}\}.
$$

By taking $k_\ell = \tau - 1, \ldots, k_2 = 1, k_1 = 0$ (where $\ell = \tau - 1$), we have proved (36) and the lemma. ∎

**Corollary 2** *Under the same condition as stated in Lemma 3, it also holds that*

$$
\left(W^{(k_\ell)} - \frac{1}{n}\mathbb{1}\mathbb{1}^T\right) \cdots \left(W^{(k_2)} - \frac{1}{n}\mathbb{1}\mathbb{1}^T\right) \left(W^{(k_1)} - \frac{1}{n}\mathbb{1}\mathbb{1}^T\right) = 0 \tag{37}
$$

*as long as* $\{\mathrm{mod}(k_1, \tau), \ldots, \mathrm{mod}(k_\ell, \tau)\} = \{0, \ldots, \tau - 1\}$.

**Proof**. Consider the production of two terms:

$$
\left(W^{(k_\ell)} - \frac{1}{n}\mathbb{1}\mathbb{1}^T\right) \left(W^{(k_{\ell-1})} - \frac{1}{n}\mathbb{1}\mathbb{1}^T\right) = W^{(k_\ell)} W^{(k_{\ell-1})} - \frac{1}{n}\mathbb{1}\mathbb{1}^T \tag{38}
$$

Here we utilize the doubly stochastic property of $W^{(k_\ell)}$ that $W^{(k_\ell)}\mathbb{1}\mathbb{1}^T = \mathbb{1}\mathbb{1}^T$. Repeating above process until all terms are merged, we

$$
\left(W^{(k_\ell)} - \frac{1}{n}\mathbb{1}\mathbb{1}^T\right) \left(W^{(k_{\ell-1})} - \frac{1}{n}\mathbb{1}\mathbb{1}^T\right) \cdots \left(W^{(k_1)} - \frac{1}{n}\mathbb{1}\mathbb{1}^T\right)
$$

$$
= W^{(k_\ell)} W^{(k_{\ell-1})} \cdots W^{(k_2)} W^{(k_1)} - \frac{1}{n}\mathbb{1}\mathbb{1}^T \tag{39}
$$

Last, referring Lemma 3, we conclude the l.h.s product in (37) is an all-zero matrix. ∎

### B.3 More about one-peer exponential graph

#### B.3.1 One-peer exponential graph with the size that is not power 2

**Numerical validation.** First, we numerically examine whether one-peer exponential graph can achieve periodic exact-averaging when the number of nodes is not the power of 2. To this end, we consider the same setting as in Fig. 4, and depict how the consensus residue $\|(\Pi_{\ell=0}^{k}W^{(\ell)} - \frac{1}{n}\mathbb{1}\mathbb{1}^{T})x\|$ decreases as iteration increases in Fig. 10. It is observed that, when $n$ is not a power of 2, one-peer exponential graphs can only achieve the asymptotic, not periodic, exact averaging.

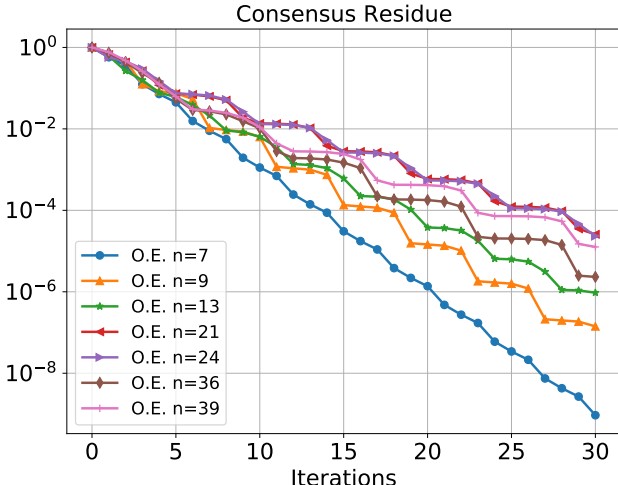

Figure 10: Illustration of how consensus residues decay with iterations for one-peer exponential graph with the size of nodes is not the power of 2.

**A case study: One-peer exponential graph with** $3$ **nodes**. We provide an example to show that it is impossible to achieve the periodic exact averaging that when the size of nodes is 3. In this case, the period is $\lceil\log_2(3)\rceil = 2$. Due to the symmetry between the nodes, the product of two one-peer exponential graph weight matrices has the form

$$
W^{(1)}W^{(0)} = \begin{bmatrix} 1-\beta & \beta & \\ & 1-\beta & \beta \\ \beta & & 1-\beta \end{bmatrix} \begin{bmatrix} 1-\alpha & & \alpha \\ \alpha & 1-\alpha & \\ & \alpha & 1-\alpha \end{bmatrix}
$$

$$
= \begin{bmatrix} 1-\alpha-\beta+2\alpha\beta & \beta-\alpha\beta & \alpha-\alpha\beta \\ \alpha-\alpha\beta & 1-\alpha-\beta+2\alpha\beta & \beta-\alpha\beta \\ \beta-\alpha\beta & \alpha-\alpha\beta & 1-\alpha-\beta+2\alpha\beta \end{bmatrix} \tag{40}
$$

In order to achieve the exact averaging, the product has to be $\frac{1}{3}\mathbb{1}_3\mathbb{1}_3^{T}$. Under this requirement, it is easy to derive that

$$
\alpha = \beta, \ \ \alpha^2 - \alpha + \frac{1}{3} = 0 \ \implies \ \alpha = \beta = \frac{1}{6}(3 \pm j\sqrt{3}) \tag{41}
$$

However, it is meaningless to let the combination weights to be complex number since the domain of iterate $x_i^{(k)}$ is $\mathbb{R}^d$.

#### B.3.2 One-peer exponential graph with uniform sampling and random permutation

In the main body, we only consider the one-peer exponential graphs in the cyclic order. However, that is not the only choice of selecting one-peer exponential graphs. Other two popular strategies are random permutation and uniform sampling. It is easy to describe these two strategies by taking an example. Consider

$$
\mathbb{W} \triangleq \{W^{(0)}, W^{(1)}, \cdots, W^{(\tau-1)}\}. \tag{42}
$$

Uniform sample strategy is at each iteration, one $W^{(t)}$ randomly selected with replacement. While random permutation is at each iteration, one $W^{(t)}$ randomly selected *without* replacement. After $\tau$ iterations, $\mathbb{W}$ will reset with $\tau$ element and repeat the sampling without replacement.

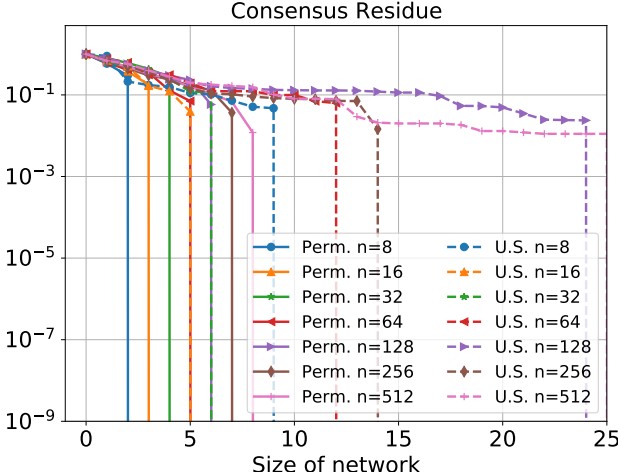

Figure 11: Illustration of how consensus residues decay with iterations for one-peer exponential graph. Perm. stands for random-permutation one-peer exponential graphs and U.S. stands for uniformed sampling one-peer exponential graphs.

With slightly modification of the proof in lemma 1, we also show that one-peer exponential graph with random permutation still has the exact averaging property. Meanwhile, one-peer exponential graph with uniform sample may no longer has this property within $\tau$-iterations. With some none-zero probability, the realization of uniform sampling with $\tau$ times can be one permutation order. Obviously, in this realization, uniform sample will have exact convergence property. Under the rest realizations, it cannot since it may miss at least one element. However, with long enough time $t$, the realization of uniform sampling with $t$ times will contain all possible elements in $\mathbb{W}$ with probability one. These claims are validated in the Fig. 11.

## C Deriving transient iteration complexity for DmSGD

We copy the convergence rate of DmSGD for non-convex costs in (3) as follows

$$\frac{1}{T}\sum_{k=1}^{T}\mathbb{E}\left\|\nabla f(\bar{\mathbf{x}}^{(k)})\right\|^2 = O\left(\frac{\sigma^2}{\sqrt{nT}} + \frac{n\sigma^2}{T(1-\rho)} + \frac{nb^2}{T(1-\rho)^2}\right) \tag{43}$$

in which $\bar{x}^{(k)} = \frac{1}{n}\sum_{i=1}^{n}x_i^{(k)}$, and the influence of momentum $\beta$ is ignored. In the following, we will derive the transient iteration complexity of DmSGD for the data-homogeneous and data-heterogeneous scenarios, respectively.

- In the **data-homogeneous** scenario, it holds that $D_i = D_j$ for any $i$ and $j$, and hence $\nabla f_i(x) = \nabla f_j(x)$. This implies that $b^2 = \frac{1}{n}\sum_{i=1}^{n}\|\nabla f_i(x) - \nabla f(x)\|^2 = 0$. To reach the linear speedup stage, the iteration $T$ has to be sufficiently large so that the $nT$-term dominates, i.e.,

$$\frac{\sigma^2}{\sqrt{nT}} \geq \frac{n\sigma^2}{T(1-\rho)}, \quad \text{which is equivalent to} \quad T \geq \frac{n^3}{(1-\rho)^2}.$$

As a result, the transient iteration complexity of DmSGD is given by $\Omega(n^3/(1-\rho)^2)$.

- In the **data-heterogeneous** scenario, it holds that $b^2 \neq 0$. To reach the linear speedup stage, the iteration $T$ has to be sufficiently large so that

$$\frac{\sigma^2}{\sqrt{nT}} \geq \frac{nb^2}{T(1-\rho)^2}, \quad \text{which is equivalent to} \quad T \geq \frac{n^3 b^4}{(1-\rho)^4 \sigma^4}.$$

As a result, the transient iteration complexity of DmSGD is given by $\Omega(n^3/(1-\rho)^4)$ if the influences of $b^4$ and $\sigma^4$ are ignored.

With the above arguments, we achieve the transient iteration complexity in (4).

# D Proof of Theorem 1

## D.1 Notations and preliminaries

**Notations.** We first introduce necessary notations as follows.

- $\mathbf{x}^{(k)} = [(\boldsymbol{x}_1^{(k)})^T; (\boldsymbol{x}_2^{(k)})^T; \cdots; (\boldsymbol{x}_n^{(k)})^T] \in \mathbb{R}^{n \times d}$

- $\mathbf{m}^{(k)} = [(\boldsymbol{m}_1^{(k)})^T; (\boldsymbol{m}_2^{(k)})^T; \cdots; (\boldsymbol{m}_n^{(k)})^T] \in \mathbb{R}^{n \times d}$

- $\nabla F(\mathbf{x}^{(k)}; \boldsymbol{\xi}^{(k)}) = [\nabla F_1(\boldsymbol{x}_1^{(k)}; \boldsymbol{\xi}_1^{(k)})^T; \cdots; \nabla F_n(\boldsymbol{x}_n^{(k)}; \boldsymbol{\xi}_n^{(k)})^T] \in \mathbb{R}^{n \times d}$

- $\nabla f(\mathbf{x}^{(k)}) = [\nabla f_1(\boldsymbol{x}_1^{(k)})^T; \nabla f_2(\boldsymbol{x}_2^{(k)})^T; \cdots; \nabla f_n(\boldsymbol{x}_n^{(k)})^T] \in \mathbb{R}^{n \times d}$

- $\bar{\mathbf{x}}^{(k)} = \left( \frac{1}{n} \sum_{i=1}^{n} \boldsymbol{x}_i^{(k)} \right)^T \in \mathbb{R}^d$

- $\bar{\mathbf{m}}^{(k)} = \left( \frac{1}{n} \sum_{i=1}^{n} \boldsymbol{m}_i^{(k)} \right)^T \in \mathbb{R}^d$

- $W = [w_{ij}] \in \mathbb{R}^{n \times n}$.

- $\mathbb{1}_n = \text{col}\{1, 1, \cdots, 1\} \in \mathbb{R}^n$

- Given two matrices $\mathbf{x}, \mathbf{y} \in \mathbb{R}^{n \times d}$, we define inner product $\langle \mathbf{x}, \mathbf{y} \rangle = \text{tr}(\mathbf{x}^T \mathbf{y})$, the Frobenius norm $\|\mathbf{x}\|^2 = \langle \mathbf{x}, \mathbf{x} \rangle$, and the $\|\mathbf{x}\|_2$ as $\mathbf{x}$'s matrix $\ell_2$ norm.

From the above definitions, it is quick to check that $\bar{\mathbf{x}}^{(k)} = \frac{1}{n} \mathbb{1}^T \mathbf{x}^{(k)}$ and $\bar{\mathbf{m}}^{(k)} = \frac{1}{n} \mathbb{1}^T \mathbf{m}^{(k)}$. We adopt the convention [4] that

$$\bar{\mathbf{x}}^{(k)} - \mathbf{x}^{(k)} \triangleq \left[ (\boldsymbol{x}_1^{(k)} - \frac{1}{n} \sum_{i=1}^{n} \boldsymbol{x}_i^{(k)})^T; (\boldsymbol{x}_2^{(k)} - \frac{1}{n} \sum_{i=1}^{n} \boldsymbol{x}_i^{(k)})^T; \cdots; (\boldsymbol{x}_n^{(k)} - \frac{1}{n} \sum_{i=1}^{n} \boldsymbol{x}_i^{(k)})^T \right] \in \mathbb{R}^{n \times d} \tag{44}$$

Same convention applies when $\bar{\mathbf{m}}^{(k)}$ adds or subtracts with the stacked variables like $\mathbf{x}^{(k)}$ and $\mathbf{m}^{(k)}$.

**Algorithm reformulation.** With the above notations, DmSGD (Algorithm 1) can be re-written as a more elegant vector-matrix form. For $k = 0, 1, \cdots$, DmSGD with one-peer exponential graph will iterate as follows:

$$\mathbf{g}^{(k)} = \nabla F(\mathbf{x}^{(k)}; \boldsymbol{\xi}^{(k)}), \tag{45}$$

$$\mathbf{m}^{(k+1)} = W^{(k)}(\beta \mathbf{m}^{(k)} + \mathbf{g}^{(k)}), \tag{46}$$

$$\mathbf{x}^{(k+1)} = W^{(k)}(\mathbf{x}^{(k)} - \gamma \mathbf{m}^{(k)}), \tag{47}$$

where $\mathbf{m}^{(0)} = 0$, $\mathbf{x}^{(0)}$ can be set arbitrarily, and $W^{(k)}$ is the weight matrix associated with the one-peer exponential graph defined by (7). Note that the weight matrix sequence $\{W^{(k)}\}$ satisfies the periodic exact averaging property, see Lemma 1.

**Smoothness.** Since each $f_i(x)$ is assumed to be $L$-smooth in Assumption A.3, it holds that $f(x) = \frac{1}{n} \sum_{i=1}^{n} f_i(x)$ is also $L$-smooth. As a result, the following inequality holds for any $\boldsymbol{x}, \boldsymbol{y} \in \mathbb{R}^d$:

$$f(\boldsymbol{x}) - f(\boldsymbol{y}) - \frac{L}{2} \|\boldsymbol{x} - \boldsymbol{y}\|^2 \leq \langle \nabla f(\boldsymbol{y}), \boldsymbol{x} - \boldsymbol{y} \rangle \tag{48}$$

**Submultiplicativity of the Frobenius norm.** Given matrices $W \in \mathbb{R}^{n \times n}$ and $\mathbf{y} \in \mathbb{R}^{n \times d}$, we have

$$\|W\mathbf{y}\| \leq \|W\|_2 \|\mathbf{y}\|. \tag{49}$$

To verify it, by letting $y_j$ be the $j$-th column of $\mathbf{y}$, we have $\|W\mathbf{y}\|^2 = \sum_{j=1}^{d} \|Wy_j\|_2^2 \leq \sum_{j=1}^{d} \|W\|_2^2 \|y_j\|_2^2 = \|W\|_2^2 \|\mathbf{y}\|^2$.

---

[4]If you are familiar with the NumPy library, it is the array broadcasting concept [https://numpy.org/doc/stable/user/basics.broadcasting.html].

**DmSGD: the averaged recursion.** Multiplying $\frac{1}{n}\mathbb{1}_n^T$ to both sides of (46) and (47), we establish the averaged (or centralized) recursion:

$$\bar{\mathbf{m}}^{(k+1)} = \beta\bar{\mathbf{m}}^{(k)} + \bar{\mathbf{g}}^{(k)} \tag{50}$$

$$\bar{\mathbf{x}}^{(k+1)} = \bar{\mathbf{x}}^{(k)} - \gamma\bar{\mathbf{m}}^{(k)} \tag{51}$$

where $\bar{\mathbf{g}}^{(k)} \triangleq \frac{1}{n}\mathbb{1}_n^T\mathbf{g}^{(k)}$.

**A critical auxiliary recursion.** We also need to introduce a auxiliary sequence $\{\bar{\mathbf{z}}^{(k)}\}$, which is commonly used for the convergence analysis in momentum methods [64, 39, 4]:

$$\bar{\mathbf{z}}^{(k)} = \frac{1}{1-\beta}(\bar{\mathbf{x}}^{(k)} - \beta\bar{\mathbf{x}}^{(k-1)}), \quad \bar{\mathbf{z}}^{(0)} = \frac{1}{\beta}\bar{\mathbf{x}}^{(0)} \tag{52}$$

It is easy to validate that [64, lemma 3]:

$$\bar{\mathbf{z}}^{(k+1)} - \bar{\mathbf{z}}^{(k)} = -\frac{\gamma}{1-\beta}\bar{\mathbf{g}}^{(k)} \tag{53}$$

When $\beta = 0$ and $\bar{\mathbf{z}}^{(0)} = \bar{\mathbf{x}}^{(0)}$, recursion (52) reduces to,

$$\bar{\mathbf{z}}^{(k)} = \bar{\mathbf{x}}^{(k)}, \quad \bar{\mathbf{x}}^{(k+1)} = \bar{\mathbf{x}}^{(k)} - \gamma\bar{\mathbf{g}}^{(k)} \tag{54}$$

**Main idea to prove Theorem 1.** Theorem 1 can be proved in two steps. First, we need to establish a decent lemma on how $f(\bar{\mathbf{z}}^{(k)})$ would evolve as iteration increases. Second, we will establish a consensus lemma showing that the consensus distance $\mathbb{E}\,\|\mathbf{x}^{(k)} - \bar{\mathbf{x}}^{(k)}\|^2$ would gradually decrease to zero. These two lemmas together will lead to the result in Theorem 1.

### D.2 Descent lemma

**Lemma 4** *Suppose the learning rate satisfies the condition* $\gamma \leq \frac{(1-\beta)^2}{2(1+\beta)L}$, *the main recursion of* (46)-(47) *under the Assumption A.1 - A.4 has*

$$\frac{1}{T+1}\sum_{k=1}^{T}\mathbb{E}\,\|\nabla f(\bar{\mathbf{x}}^{(k)})\|^2 \leq \frac{2(1-\beta)}{\gamma(T+1)}(\mathbb{E}\,f(\bar{\mathbf{z}}^{(0)}) - f^\star) + \frac{\gamma L}{n(1-\beta)}\sigma^2 + \frac{\beta L\gamma}{n(1-\beta)^2}\sigma^2$$

$$+ \frac{L^2}{T+1}\sum_{k=0}^{T}\mathbb{E}\,\|\bar{\mathbf{x}}^{(k)} - \mathbf{x}^{(k)}\|^2 \tag{55}$$

*where* $f^\star$ *is the minimum value of the problem* (1)*;* $\sigma^2$ *and L are the constants defined in Assumptions.*

**Proof**. Utilizing the $L$-smooth assumption of loss function $f$ – Eq. (48), we have:

$$\mathbb{E}\,f(\bar{\mathbf{z}}^{(k+1)})$$

$$\leq \mathbb{E}\,f(\bar{\mathbf{z}}^{(k)}) + \mathbb{E}\,\langle\bar{\mathbf{z}}^{(k+1)} - \bar{\mathbf{z}}^{(k)}, \nabla f(\bar{\mathbf{z}}^{(k)})\rangle + \frac{L}{2}\mathbb{E}\,\|\bar{\mathbf{z}}^{(k+1)} - \bar{\mathbf{z}}^{(k)}\|^2$$

$$\overset{(a)}{=} \mathbb{E}\,f(\bar{\mathbf{z}}^{(k)}) - \frac{\gamma}{1-\beta}\mathbb{E}\,\langle\bar{\mathbf{g}}^{(k)}, \nabla f(\bar{\mathbf{z}}^{(k)})\rangle + \frac{\gamma^2 L}{2(1-\beta)^2}\mathbb{E}\,\|\bar{\mathbf{g}}^{(k)}\|^2$$

$$\overset{(b)}{=} \mathbb{E}\,f(\bar{\mathbf{z}}^{(k)}) - \frac{\gamma}{1-\beta}\mathbb{E}\,\left\langle\frac{1}{n}\sum_{i=1}^{n}\nabla f_i(\boldsymbol{x}_i^{(k)}), \nabla f(\bar{\mathbf{z}}^{(k)})\right\rangle + \frac{\gamma^2 L}{2(1-\beta)^2}\mathbb{E}\,\|\bar{\mathbf{g}}^{(k)}\|^2 \tag{56}$$

where step (a) expands $\bar{\mathbf{z}}^{(k+1)} - \bar{\mathbf{z}}^{(k)}$ according to (53) and step (b) utilizes the unbiased and independent assumption of gradient noise (Assumption A.2):

$$\mathbb{E}\,\langle\bar{\mathbf{g}}^{(k)}, \nabla f(\bar{\mathbf{z}}^{(k)})\rangle$$

$$= \mathbb{E}\,\left\langle\frac{1}{n}\sum_{i=1}^{n}\nabla F_i(\boldsymbol{x}_i^{(k)};\boldsymbol{\xi}_i) - \frac{1}{n}\sum_{i=1}^{n}\nabla f_i(\boldsymbol{x}_i^{(k)}) + \frac{1}{n}\sum_{i=1}^{n}\nabla f_i(\boldsymbol{x}_i^{(k)}), \nabla f(\bar{\mathbf{z}}^{(k)})\right\rangle$$

$$= \mathbb{E}\,\left\langle\frac{1}{n}\sum_{i=1}^{n}\nabla F_i(\boldsymbol{x}_i^{(k)};\boldsymbol{\xi}_i) - \frac{1}{n}\sum_{i=1}^{n}\nabla f_i(\boldsymbol{x}_i^{(k)}), \nabla f(\bar{\mathbf{z}}^{(k)})\right\rangle + \mathbb{E}\,\left\langle\frac{1}{n}\sum_{i=1}^{n}\nabla f_i(\boldsymbol{x}_i^{(k)}), \nabla f(\bar{\mathbf{z}}^{(k)})\right\rangle$$

$$= \mathbb{E}\,\left\langle\frac{1}{n}\sum_{i=1}^{n}\nabla f_i(\boldsymbol{x}_i^{(k)}), \nabla f(\bar{\mathbf{z}}^{(k)})\right\rangle \tag{57}$$

Next, we focus on bounding the middle term in (56). First, we expand it into:

$$-\mathbb{E}\left\langle \frac{1}{n}\sum_{i=1}^{n}\nabla f_i(\boldsymbol{x}_i^{(k)}), \nabla f(\bar{\mathbf{z}}^{(k)})\right\rangle \tag{58}$$

$$= -\mathbb{E}\left(\frac{1}{n}\sum_{i=1}^{n}\nabla f_i(\boldsymbol{x}_i^{(k)})\right)^T (\nabla f(\bar{\mathbf{z}}^{(k)}) - \nabla f(\bar{\mathbf{x}}^{(k)})) - \mathbb{E}\left(\frac{1}{n}\sum_{i=1}^{n}\nabla f_i(\boldsymbol{x}_i^{(k)})\right)^T \nabla f(\bar{\mathbf{x}}^{(k)})$$

To bound the first term in (58), we apply the Young's inequality $-a^T b \leq \frac{1}{2\epsilon}\|a\|^2 + \frac{\epsilon}{2}\|b\|^2$:

$$-\left(\frac{1}{n}\sum_{i=1}^{n}\nabla f_i(\boldsymbol{x}_i^{(k)})\right)^T (\nabla f(\bar{\mathbf{z}}^{(k)}) - \nabla f(\bar{\mathbf{x}}^{(k)}))$$

$$\leq \frac{1}{2\epsilon}\|\frac{1}{n}\sum_{i=1}^{n}\nabla f_i(\boldsymbol{x}_i^{(k)})\|^2 + \frac{\epsilon}{2}\|\nabla f(\bar{\mathbf{z}}^{(k)}) - \nabla f(\bar{\mathbf{x}}^{(k)})\|^2$$

$$\leq \frac{1}{2\epsilon}\|\frac{1}{n}\sum_{i=1}^{n}\nabla f_i(\boldsymbol{x}_i^{(k)})\|^2 + \frac{\epsilon L^2}{2}\|\bar{\mathbf{z}}^{(k)} - \bar{\mathbf{x}}^{(k)}\|^2$$

$$= \frac{1}{2\epsilon}\|\frac{1}{n}\sum_{i=1}^{n}\nabla f_i(\boldsymbol{x}_i^{(k)})\|^2 + \frac{\epsilon L^2 \beta^2 \gamma^2}{2(1-\beta)^2}\|\bar{\mathbf{m}}^{(k)}\|^2 \tag{59}$$

where the last equality relied on the observation that

$$\bar{\mathbf{z}}^{(k)} - \bar{\mathbf{x}}^{(k)} = \frac{\beta}{1-\beta}[\bar{\mathbf{x}}^{(k)} - \bar{\mathbf{x}}^{(k-1)}] = -\frac{\beta\gamma}{1-\beta}\bar{\mathbf{m}}^{(k)} \tag{60}$$

If we choose $\epsilon = \frac{(1-\beta)^2}{\gamma\beta L}$, we obtain:

$$-\left(\frac{1}{n}\sum_{i=1}^{n}\nabla f_i(\boldsymbol{x}_i^{(k)})\right)^T (\nabla f(\bar{\mathbf{z}}^{(k)}) - \nabla f(\bar{\mathbf{x}}^{(k)})) \leq \frac{\beta L\gamma}{2(1-\beta)^2}\|\frac{1}{n}\sum_{i=1}^{n}\nabla f_i(\boldsymbol{x}_i^{(k)})\|^2 + \frac{\beta L\gamma}{2}\|\bar{\mathbf{m}}^{(k)}\|^2$$

$$\tag{61}$$

To bound the second term in (58), we use the identity that $a^T b = \frac{1}{2}\left(\|a\|^2 + \|b\|^2 - \|a-b\|^2\right)$:

$$\left(\frac{1}{n}\sum_{i=1}^{n}\nabla f_i(\boldsymbol{x}_i^{(k)})\right)^T \nabla f(\bar{\mathbf{x}}^{(k)})$$

$$= \frac{1}{2}\left(\|\nabla f(\bar{\mathbf{x}}^{(k)})\|\|^2 + \|\frac{1}{n}\sum_{i=1}^{n}\nabla f_i(\boldsymbol{x}_i^{(k)})\|^2 - \|\nabla f(\bar{\mathbf{x}}^{(k)}) - \frac{1}{n}\sum_{i=1}^{n}f_i(\boldsymbol{x}_i^{(k)})\|^2\right)$$

$$\geq \frac{1}{2}\left(\|\nabla f(\bar{\mathbf{x}}^{(k)})\|\|^2 + \|\frac{1}{n}\sum_{i=1}^{n}\nabla f_i(\boldsymbol{x}_i^{(k)})\|^2 - \frac{L^2}{n}\sum_{i=1}^{n}\|\bar{\mathbf{x}}^{(k)} - \boldsymbol{x}_i^{(k)}\|^2\right)$$

$$= \frac{1}{2}\left(\|\nabla f(\bar{\mathbf{x}}^{(k)})\|\|^2 + \|\frac{1}{n}\sum_{i=1}^{n}\nabla f_i(\boldsymbol{x}_i^{(k)})\|^2 - L^2\|\bar{\mathbf{x}}^{(k)} - \mathbf{x}^{(k)}\|^2\right) \tag{62}$$

Noting (61) and (62) hold for all realization. Substituting them back to (58) and simplifying the terms, the middle term in (56) is bounded as follows

$$
-\frac{\gamma}{1-\beta}\mathbb{E}\left(\frac{1}{n}\sum_{i=1}^{n}\nabla f_i(\boldsymbol{x}_i^{(k)})\right)^T(\nabla f(\bar{\mathbf{z}}^{(k)}))
$$

$$
\leq \frac{\gamma^2\beta L}{2(1-\beta)^3}\mathbb{E}\,\|\frac{1}{n}\sum_{i=1}^{n}\nabla f_i(\boldsymbol{x}_i^{(k)})\|^2 + \frac{\beta L\gamma^2}{2(1-\beta)}\mathbb{E}\,\|\bar{\mathbf{m}}^{(k)}\|^2 + \frac{L^2\gamma}{2(1-\beta)n}\sum_{i=1}^{n}\mathbb{E}\,\|\bar{\mathbf{x}}^{(k)}-\boldsymbol{x}_i^{(k)}\|^2
$$

$$
-\frac{\gamma}{2(1-\beta)}\mathbb{E}\,\|\nabla f(\bar{\mathbf{x}}^{(k)})\|^2 - \frac{\gamma}{2(1-\beta)}\mathbb{E}\,\|\frac{1}{n}\sum_{i=1}^{n}\nabla f_i(\boldsymbol{x}_i^{(k)})\|^2
$$

$$
=\frac{\beta L\gamma^2}{2(1-\beta)}\mathbb{E}\,\|\bar{\mathbf{m}}^{(k)}\|^2 + \frac{L^2\gamma}{2(1-\beta)}\mathbb{E}\,\|\bar{\mathbf{x}}^{(k)}-\mathbf{x}^{(k)}\|^2
$$

$$
-\frac{\gamma}{2(1-\beta)}\mathbb{E}\,\|\nabla f(\bar{\mathbf{x}}^{(k)})\|^2 - \frac{\gamma}{2(1-\beta)}\left(1-\frac{\gamma\beta L}{(1-\beta)^2}\right)\mathbb{E}\,\|\frac{1}{n}\sum_{i=1}^{n}\nabla f_i(\boldsymbol{x}_i^{(k)})\|^2 \tag{63}
$$

The third term in the main recursion (56) can be bounded as similar as we did in (57):

$$
\mathbb{E}\,\|\bar{\mathbf{g}}^{(k)}\|^2 = \mathbb{E}\,\left\|\frac{1}{n}\sum_{i=1}^{n}\nabla f_i(\boldsymbol{x}_i^{(k)}) + \frac{1}{n}\sum_{i=1}^{n}\nabla F_i(\boldsymbol{x}_i^{(k)};\boldsymbol{\xi}_i) - \frac{1}{n}\sum_{i=1}^{n}\nabla f_i(\boldsymbol{x}_i^{(k)})\right\|^2
$$

$$
\overset{(a)}{=}\mathbb{E}\,\left\|\frac{1}{n}\sum_{i=1}^{n}\nabla f_i(\boldsymbol{x}_i^{(k)})\right\|^2 + \mathbb{E}\,\left\|\frac{1}{n}\sum_{i=1}^{n}\nabla F_i(\boldsymbol{x}_i^{(k)};\boldsymbol{\xi}_i) - \frac{1}{n}\sum_{i=1}^{n}\nabla f_i(\boldsymbol{x}_i^{(k)})\right\|^2
$$

$$
\overset{(b)}{=}\mathbb{E}\,\left\|\frac{1}{n}\sum_{i=1}^{n}\nabla f_i(\boldsymbol{x}_i^{(k)})\right\|^2 + \frac{1}{n^2}\sum_{i=1}^{n}\mathbb{E}\,\left\|\nabla F_i(\boldsymbol{x}_i^{(k)};\boldsymbol{\xi}_i) - \nabla f_i(\boldsymbol{x}_i^{(k)})\right\|^2
$$

$$
\leq \mathbb{E}\,\left\|\frac{1}{n}\sum_{i=1}^{n}\nabla f_i(\boldsymbol{x}_i^{(k)})\right\|^2 + \frac{1}{n}\sigma^2 \tag{64}
$$

where the step (a) that separates the norm square term into the sum of two terms is thanks to the independent assumption of gradient noise over the past data; the step (b) is one of the key step that relied on the independent assumption of gradient noise across the agents. Substituting (63) and (64) back to main recursion (56) and re-organize the terms, we establish

$$
\frac{\gamma}{2(1-\beta)}\mathbb{E}\,\|\nabla f(\bar{\mathbf{x}}^{(k)})\|^2 \leq \mathbb{E}\,f(\bar{\mathbf{z}}^{(k)}) - \mathbb{E}\,f(\bar{\mathbf{z}}^{(k+1)}) + \frac{\gamma^2 L}{2(1-\beta)^2}\mathbb{E}\,\left\|\frac{1}{n}\sum_{i=1}^{n}\nabla f_i(\boldsymbol{x}_i^{(k)})\right\|^2
$$

$$
+\frac{\beta L\gamma^2}{2(1-\beta)}\mathbb{E}\,\|\bar{\mathbf{m}}^{(k)}\|^2 + \frac{L^2\gamma}{2(1-\beta)}\mathbb{E}\,\|\bar{\mathbf{x}}^{(k)}-\mathbf{x}^{(k)}\|^2
$$

$$
-\frac{\gamma}{2(1-\beta)}\left(1-\frac{\gamma\beta L}{(1-\beta)^2}\right)\mathbb{E}\,\|\frac{1}{n}\sum_{i=1}^{n}\nabla f_i(\boldsymbol{x}_i^{(k)})\|^2 + \frac{\gamma^2 L}{2(1-\beta)^2 n}\sigma^2 \tag{65}
$$

Next step is to expand the momentum term $\mathbb{E}\,\|\bar{\mathbf{m}}^{(k)}\|^2$ back to the first iteration:

$$
\bar{\mathbf{m}}^{(k)} = \sum_{t=0}^{k-1}\beta^{k-1-t}\bar{\mathbf{g}}^{(t)} \tag{66}
$$

Taking the expectation and norm square on both sides, we have

$$
\begin{aligned}
\mathbb{E}\left\|\bar{\mathbf{m}}^{(k)}\right\|^2 &\overset{(a)}{\leq} \mathbb{E}\left\|\frac{s_k}{s_k}\sum_{t=0}^{k-1}\beta^{k-1-t}\bar{\mathbf{g}}^{(t)}\right\|^2 \\
&\overset{(b)}{\leq} s_k\sum_{t=0}^{k-1}\beta^{k-1-t}\mathbb{E}\left\|\bar{\mathbf{g}}^{(t)}\right\|^2 \\
&\overset{(c)}{\leq} s_k\sum_{t=0}^{k-1}\beta^{k-1-t}\mathbb{E}\left\|\frac{1}{n}\sum_{i=1}^{n}\nabla f_i(\boldsymbol{x}_i^{(t)})\right\|^2 + \frac{s_k^2}{n}\sigma^2
\end{aligned}
\tag{67}
$$

where in step (a), we define $s_k \triangleq \sum_{t=0}^{k-1}\beta^{k-1-t}$ as the sum of weights; we applied the Jensen's inequality in the step (b); step (c) used the conclusion from (64). Note the sum of weight $s_k$ is bounded by constant:

$$
s_k = \frac{1-\beta^k}{1-\beta} \leq \frac{1}{1-\beta}
\tag{68}
$$

Plug it back to (65), we have

$$
\begin{aligned}
\frac{\gamma}{2(1-\beta)}\mathbb{E}\left\|\nabla f(\bar{\mathbf{x}}^{(k)})\right\|^2 \leq{}& \mathbb{E}\,f(\bar{\mathbf{z}}^{(k)}) - \mathbb{E}\,f(\bar{\mathbf{z}}^{(k+1)}) + \frac{\gamma^2 L}{2n(1-\beta)^2}\sigma^2 + \frac{s_k^2\beta L\gamma^2}{2n(1-\beta)}\sigma^2 \\
&+ \frac{s_k\beta L\gamma^2}{2(1-\beta)}\sum_{t=0}^{k-1}\beta^{k-1-t}\mathbb{E}\left\|\frac{1}{n}\sum_{i=1}^{n}\nabla f_i(\boldsymbol{x}_i^{(t)})\right\|^2 \\
&+ \frac{L^2\gamma}{2(1-\beta)}\mathbb{E}\left\|\bar{\mathbf{x}}^{(k)}-\mathbf{x}^{(k)}\right\|^2 \\
&- \frac{\gamma}{2(1-\beta)}\left(1-\frac{\gamma L}{1-\beta}-\frac{\gamma\beta L}{(1-\beta)^2}\right)\mathbb{E}\|\frac{1}{n}\sum_{i=1}^{n}\nabla f_i(\boldsymbol{x}_i^{(k)})\|^2
\end{aligned}
\tag{69}
$$

Taking the average of (69) from time $k=0$ to $k=T$, we have

$$
\begin{aligned}
\frac{\gamma}{2(T+1)(1-\beta)}&\sum_{k=0}^{T}\mathbb{E}\left\|\nabla f(\bar{\mathbf{x}}^{(k)})\right\|^2 \\
\leq{}& \frac{1}{T+1}\left(\mathbb{E}\,f(\bar{\mathbf{z}}^{(0)}) - \mathbb{E}\,f(\bar{\mathbf{z}}^{(T+1)})\right) + \frac{\gamma^2 L}{2n(1-\beta)^2}\sigma^2 + \frac{\beta L\gamma^2}{2n(1-\beta)^3}\sigma^2 \\
&+ \frac{1}{T+1}\sum_{k=0}^{T}\frac{s_k\beta L\gamma^2}{2(1-\beta)}\sum_{t=0}^{k-1}\beta^{k-1-t}\mathbb{E}\left\|\frac{1}{n}\sum_{i=1}^{n}\nabla f_i(\boldsymbol{x}_i^{(t)})\right\|^2 \\
&+ \frac{L^2\gamma}{2(1-\beta)}\frac{1}{T+1}\sum_{k=0}^{T}\mathbb{E}\left\|\bar{\mathbf{x}}^{(k)}-\mathbf{x}^{(k)}\right\|^2 \\
&- \frac{\gamma}{2(1-\beta)}\left(1-\frac{\gamma L}{1-\beta}-\frac{\gamma\beta L}{(1-\beta)^2}\right)\frac{1}{T+1}\sum_{k=0}^{T}\mathbb{E}\|\frac{1}{n}\sum_{i=1}^{n}\nabla f_i(\boldsymbol{x}_i^{(k)})\|^2
\end{aligned}
\tag{70}
$$

Focus on the term in the second line of r.h.s of (70):

$$\frac{1}{T+1}\sum_{k=0}^{T}\frac{s_k\beta L\gamma^2}{2(1-\beta)}\sum_{t=0}^{k-1}\beta^{k-1-t}\mathbb{E}\left\|\frac{1}{n}\sum_{i=1}^{n}\nabla f_i(\boldsymbol{x}_i^{(t)})\right\|^2$$

$$\overset{(a)}{\leq}\frac{\beta L\gamma^2}{2(1-\beta)^2}\frac{1}{T+1}\sum_{k=0}^{T}\sum_{t=0}^{k-1}\beta^{k-1-t}\mathbb{E}\left\|\frac{1}{n}\sum_{i=1}^{n}\nabla f_i(\boldsymbol{x}_i^{(t)})\right\|^2$$

$$\overset{(b)}{=}\frac{\beta L\gamma^2}{2(1-\beta)^2}\frac{1}{T+1}\sum_{t=0}^{T-1}\sum_{k=t+1}^{T}\beta^{k-1-t}\mathbb{E}\left\|\frac{1}{n}\sum_{i=1}^{n}\nabla f_i(\boldsymbol{x}_i^{(t)})\right\|^2$$

$$\overset{(c)}{\leq}\frac{\beta L\gamma^2}{2(1-\beta)^3}\frac{1}{T+1}\sum_{k=0}^{T}\mathbb{E}\left\|\frac{1}{n}\sum_{i=1}^{n}\nabla f_i(\boldsymbol{x}_i^{(k)})\right\|^2 \tag{71}$$

where step (a) uses the upper bound of $s_k$ in (68); step (b) switches the order of two summations; step (c), again, uses the upper bound of $s_k$ and re-align the index of summation due to non-negativity of each term. Hence, we establish

$$\frac{\gamma}{2(T+1)(1-\beta)}\sum_{k=1}^{T}\mathbb{E}\left\|\nabla f(\bar{\mathbf{x}}^{(k)})\right\|^2$$

$$\leq\frac{1}{T+1}\left(\mathbb{E}f(\bar{\mathbf{z}}^{(0)})-\mathbb{E}f(\bar{\mathbf{z}}^{(T+1)})\right)+\frac{\gamma^2 L}{2n(1-\beta)^2}\sigma^2+\frac{\beta L\gamma^2}{2n(1-\beta)^3}\sigma^2$$

$$+\frac{L^2\gamma}{2(1-\beta)}\frac{1}{T+1}\sum_{k=0}^{T}\mathbb{E}\left\|\bar{\mathbf{x}}^{(k)}-\mathbf{x}^{(k)}\right\|^2$$

$$-\frac{\gamma}{2(1-\beta)}\left(1-\frac{\gamma L}{1-\beta}-\frac{2\gamma\beta L}{(1-\beta)^2}\right)\frac{1}{T+1}\sum_{k=0}^{T}\mathbb{E}\|\frac{1}{n}\sum_{i=1}^{n}\nabla f_i(\boldsymbol{x}_i^{(k)})\|^2 \tag{72}$$

In order to discard the $\mathbb{E}\|\frac{1}{n}\sum_{i=1}^{n}\nabla f_i(\boldsymbol{x}_i^{(k)})\|^2$, the step-size has to be small enough so that the coefficient is negative. To achieve that, we need $1-\frac{\gamma L}{1-\beta}-\frac{2\gamma\beta L}{(1-\beta)^2}\geq 0$. The idea is we can require last two terms bounded by two constants, which sum up to 1. Suppose we require that:

$$\frac{\gamma L}{1-\beta}\leq\frac{1-\beta}{1-\beta^2}\implies\gamma\leq\frac{1-\beta}{(1+\beta)L} \tag{73}$$

$$\frac{2\gamma\beta L}{(1-\beta)^2}\leq\frac{\beta(1-\beta)}{1-\beta^2}\implies\gamma\leq\frac{(1-\beta)^2}{2(1+\beta)L} \tag{74}$$

Since $1>\beta\geq 0$, (74) is always smaller than (73). So as long as $\gamma\leq\frac{(1-\beta)^2}{2(1+\beta)L}$, we can safely discard the last terms in (72).

Finally, we arrive at the conclusion in the lemma by noting $f^\star$ is the minimum value of the problem:

$$\frac{1}{T+1}\sum_{k=0}^{T}\mathbb{E}\left\|\nabla f(\bar{\mathbf{x}}^{(k)})\right\|^2\leq\frac{2(1-\beta)}{\gamma(T+1)}(\mathbb{E}f(\bar{\mathbf{z}}^{(0)})-f^\star)+\frac{\gamma L}{n(1-\beta)}\sigma^2+\frac{\beta L\gamma}{n(1-\beta)^2}\sigma^2$$

$$+\frac{L^2}{T+1}\sum_{k=0}^{T}\mathbb{E}\left\|\bar{\mathbf{x}}^{(k)}-\mathbf{x}^{(k)}\right\|^2 \tag{75}$$

A few comments about this bounds: the historical average of gradient at the average trajectory $\bar{\mathbf{x}}^{(k)}$ is bounded by the excess risk at the initial value, the gradient noise, and the average of the consensus residue over the time. ∎

### D.3  Consensus lemma

Before we can bound the consensus residue of the DmSGD algorithm, we transform the main recursion (46) and (47) into the following consensus residue form, which is much easier for analysis.

Because of the periodic exact averaging property, we can view the main recursion in every $\tau$ iterations as reference point. Recall $\tau = \ln(n)$ which is an integer. We define $m = \lfloor k/\tau \rfloor - 1$. (More precisely, $m$ should be a function of $k$. $m(k)$ would be more proper but we choose $m$ to light the notation). Apparently, it holds that $2\tau > k - m\tau \geq \tau$. It implies from iteration $k$ to $m\tau$ it must contain as least one period.

**Lemma 5** *If we expand the recursion from iteration $k$ to the previous period $m\tau$, it has following concise form due to exact averaging property:*

$$\mathbf{x}^{(k)} - \bar{\mathbf{x}}^{(k)} = -\gamma \sum_{t=m\tau}^{k-1} \left( \sum_{j=t+1}^{k-1} \beta^{j-1-t} \right) \left( \prod_{i=t}^{k-1} \widehat{W}^{(i)} \right) (\mathbf{g}^{(t)} - \bar{\mathbf{g}}^{(t)}), \quad \forall k \geq \tau \tag{76}$$

*where $\widehat{W}^{(i)} \triangleq W^{(i)} - \frac{1}{n}\mathbf{1}\mathbf{1}^T$.*

**Proof**: Recalling that decentralized momentum SGD in (46) subtract it by the centralized recursion:

$$\mathbf{x}^{(k)} - \bar{\mathbf{x}}^{(k)} = W^{(k-1)}(\mathbf{x}^{(k-1)} - \bar{\mathbf{x}}^{(k-1)} - \gamma(\mathbf{m}^{(k-1)} - \bar{\mathbf{m}}^{(k-1)}))$$

$$= \left( W^{(k-1)} - \frac{1}{n}\mathbf{1}\mathbf{1}^T \right) (\mathbf{x}^{(k-1)} - \bar{\mathbf{x}}^{(k-1)} - \gamma(\mathbf{m}^{(k-1)} - \bar{\mathbf{m}}^{(k-1)})) \tag{77}$$

where we utilized the average of the average value is still the average value: $W^{(k-1)}\bar{\mathbf{x}}^{(k-1)} = \bar{\mathbf{x}}^{(k-1)}$ and $W^{(k-1)}\bar{\mathbf{m}}^{(k-1)} = \bar{\mathbf{m}}^{(k-1)}$. For the short notation, we denote that

$$\widehat{W}^{(k-1)} := W^{(k-1)} - \frac{1}{n}\mathbf{1}\mathbf{1}^T \tag{78}$$

For any $k \geq \tau$, we can always expand the recursion into $m\tau$:

$$\mathbf{x}^{(k)} - \bar{\mathbf{x}}^{(k)} = \widehat{W}^{(k-1)}(\mathbf{x}^{(k-1)} - \bar{\mathbf{x}}^{(k-1)} - \gamma(\mathbf{m}^{(k-1)} - \bar{\mathbf{m}}^{(k-1)}))$$

$$= \prod_{i=m\tau}^{k-1} \widehat{W}^{(i)}(\mathbf{x}^{(m\tau)} - \bar{\mathbf{x}}^{(m\tau)}) - \gamma \sum_{j=m\tau}^{k-1} \prod_{i=j}^{k-1} \widehat{W}^{(i)}(\mathbf{m}^{(j)} - \bar{\mathbf{m}}^{(j)})$$

$$\overset{(a)}{=} -\gamma \sum_{j=m\tau}^{k-1} \prod_{i=j}^{k-1} \widehat{W}^{(i)}(\mathbf{m}^{(j)} - \bar{\mathbf{m}}^{(j)})$$

$$= -\gamma \sum_{j=m\tau}^{k-1} \prod_{i=j}^{k-1} W^{(i)}(\mathbf{m}^{(j)} - \bar{\mathbf{m}}^{(j)}) \tag{79}$$

where step (a) discards the first term because of the periodic exact averaging property in Lemma 1. To evaluate the sum of production term in (79), we first expand the momentum term according the recursion (46) until iteration $m\tau$

$$\mathbf{m}^{(j)} = \beta^{j-m\tau} \prod_{i=m\tau}^{j-1} W^{(i)}\mathbf{m}^{(m\tau)} + \sum_{t=m\tau}^{j-1} \beta^{j-1-t} \prod_{q=t}^{j-1} W^{(q)}\mathbf{g}^{(t)} \tag{80}$$

Multiplying $\prod_{i=j}^{k-1} W^{(i)}$ on both sides and note we can exchange the order of $\prod_i$ and $\sum_t$ when their index is not dependent:

$$\prod_{i=j}^{k-1} W^{(i)}\mathbf{m}^{(j)} = \beta^{j-m\tau} \prod_{i=j}^{k-1} W^{(i)} \prod_{i=m\tau}^{j-1} W^{(i)}\mathbf{m}^{(m\tau)} + \sum_{t=m\tau}^{j-1} \beta^{j-1-t} \prod_{i=j}^{k-1} W^{(i)} \prod_{q=t}^{j-1} W^{(q)}\mathbf{g}^{(t)}$$

$$= \beta^{j-m\tau} \prod_{i=m\tau}^{k-1} W^{(i)}\mathbf{m}^{(m\tau)} + \sum_{t=m\tau}^{j-1} \beta^{j-1-t} \prod_{i=t}^{k-1} W^{(i)}\mathbf{g}^{(t)}$$

$$= \beta^{j-m\tau}\bar{\mathbf{m}}^{(m\tau)} + \sum_{t=m\tau}^{j-1} \beta^{j-1-t} \prod_{i=t}^{k-1} W^{(i)}\mathbf{g}^{(t)} \tag{81}$$

where the last equality is, again, thanks to the periodic exact averaging property. We can establish the similar conclusion for average momentum term:

$$\prod_{i=j}^{k-1} W^{(i)} \bar{\mathbf{m}}^{(j)} = \bar{\mathbf{m}}^{(j)} = \beta^{j-m\tau} \bar{\mathbf{m}}^{(m\tau)} + \sum_{t=m\tau}^{j-1} \beta^{j-1-t} \bar{\mathbf{g}}^{(t)}$$

$$= \beta^{j-m\tau} \bar{\mathbf{m}}^{(m\tau)} + \sum_{t=m\tau}^{j-1} \beta^{j-1-t} \prod_{i=t}^{k-1} W^{(i)} \bar{\mathbf{g}}^{(t)} \qquad (82)$$

Combining above two, we get

$$\prod_{i=j}^{k-1} W^{(i)} (\mathbf{m}^{(j)} - \bar{\mathbf{m}}^{(j)}) = \sum_{t=m\tau}^{j-1} \beta^{j-1-t} \prod_{i=t}^{k-1} W^{(i)} (\mathbf{g}^{(t)} - \bar{\mathbf{g}}^{(t)})$$

$$= \sum_{t=m\tau}^{j-1} \beta^{j-1-t} \prod_{i=t}^{k-1} \widehat{W}^{(i)} (\mathbf{g}^{(t)} - \bar{\mathbf{g}}^{(t)}) \qquad (83)$$

Substituting (83) back to (79), we establish

$$\mathbf{x}^{(k)} - \bar{\mathbf{x}}^{(k)} = -\gamma \sum_{j=m\tau}^{k-1} \sum_{t=m\tau}^{j-1} \beta^{j-1-t} \prod_{i=t}^{k-1} \widehat{W}^{(i)} (\mathbf{g}^{(t)} - \bar{\mathbf{g}}^{(t)}) \qquad (84)$$

Note we can switch the order of two summations:

$$\sum_{j=m\tau}^{k-1} \sum_{t=m\tau}^{j-1} \equiv \sum_{t=m\tau}^{k-1} \sum_{j=t+1}^{k-1} \qquad (85)$$

By above identity, we can group the coefficients and finally arrive at

$$\mathbf{x}^{(k)} - \bar{\mathbf{x}}^{(k)} = -\gamma \sum_{t=m\tau}^{k-1} \left( \sum_{j=t+1}^{k-1} \beta^{j-1-t} \right) \left( \prod_{i=t}^{k-1} \widehat{W}^{(i)} \right) (\mathbf{g}^{(t)} - \bar{\mathbf{g}}^{(t)}) \qquad (86)$$

$\blacksquare$

With this simplified consensus residue form (86), we ready to present the consensus lemma.

**Lemma 6 (Consensus Lemma)** *Suppose the learning rate satisfies the condition $\gamma \leq \frac{1-\beta}{6L\tau}$ and Assumption A.1 - A.4 holds, the consensus residue have*

$$\frac{1}{T+1} \sum_{k=0}^{T} \mathbb{E} \left\| \mathbf{x}^{(k)} - \bar{\mathbf{x}}^{(k)} \right\|^2 \leq \frac{8\tau\gamma^2\rho_{\max}^2}{(1-\beta)^2}(\sigma^2 + 4\tau b^2) + \frac{2}{(T+1)} \sum_{k=0}^{\tau-1} \mathbb{E} \left\| \mathbf{x}^{(k)} - \bar{\mathbf{x}}^{(k)} \right\|^2 \quad (87)$$

*where $\sigma^2$ and $b^2$ are the constants defined in Assumptions for gradient noise and data heterogeneous respectively; the spectral gap $\rho_{\max}^2$ is defined as*

$$\rho_{\max}^2 = \max_{i \in [0,\tau-1]} \left\| \widehat{W}^{(i)} \right\|_2^2 \leq 1 \qquad (88)$$

**Proof**. Taking norm and expectation on both sides of (86), we obtain

$$\mathbb{E} \left\| \mathbf{x}^{(k)} - \bar{\mathbf{x}}^{(k)} \right\|^2$$

$$= \gamma^2 \mathbb{E} \left\| \sum_{t=m\tau}^{k-1} \left( \sum_{j=t+1}^{k-1} \beta^{j-1-t} \right) \left( \prod_{i=t}^{k-1} \widehat{W}^{(i)} \right) (\mathbf{g}^{(t)} - \bar{\mathbf{g}}^{(t)}) \right\|^2$$

$$\leq 2\gamma^2 \mathbb{E} \underbrace{\left\| \sum_{t=m\tau}^{k-1} \left( \sum_{j=t+1}^{k-1} \beta^{j-1-t} \right) \left( \prod_{i=t}^{k-1} \widehat{W}^{(i)} \right) (\nabla F(\mathbf{x}^{(t)}) - \nabla f(\mathbf{x}^{(t)})) \right\|^2}_{:=(A)}$$

$$+ 2\gamma^2 \mathbb{E} \underbrace{\left\| \sum_{t=m\tau}^{k-1} \left( \sum_{j=t+1}^{k-1} \beta^{j-1-t} \right) \left( \prod_{i=t}^{k-1} \widehat{W}^{(i)} \right) (\mathbf{g}^{(t)} - \bar{\mathbf{g}}^{(t)} - \nabla F(\mathbf{x}^{(t)}) + \nabla f(\mathbf{x}^{(t)})) \right\|^2}_{:=(B)} \quad (89)$$

where the inequality is due to Jensen's inequality. First, let's exam the second term in (89), which contains the gradient noise only

$$(B) \overset{(a)}{=} 2\gamma^2 \sum_{t=m\tau}^{k-1} \mathbb{E} \left\| \left( \sum_{j=t+1}^{k-1} \beta^{j-1-t} \right) \left( \prod_{i=t}^{k-1} \widehat{W}^{(i)} \right) (\mathbf{g}^{(t)} - \bar{\mathbf{g}}^{(t)} - \nabla F(\mathbf{x}^{(t)}) + \nabla f(\mathbf{x}^{(t)})) \right\|^2$$

$$\overset{(b)}{\leq} \frac{2\gamma^2}{1-\beta} \sum_{t=m\tau}^{k-1} \sum_{j=t+1}^{k-1} \beta^{j-1-t} \mathbb{E} \left\| \left( \prod_{i=t}^{k-1} \widehat{W}^{(i)} \right) (\mathbf{g}^{(t)} - \bar{\mathbf{g}}^{(t)} - \nabla F(\mathbf{x}^{(t)}) + \nabla f(\mathbf{x}^{(t)})) \right\|^2$$

$$\overset{(c)}{\leq} \frac{2\gamma^2}{1-\beta} \sum_{t=m\tau}^{k-1} \sum_{j=t+1}^{k-1} \beta^{j-1-t} \left\| \prod_{i=t}^{k-1} \widehat{W}^{(i)} \right\|_2^2 \mathbb{E} \left\| \mathbf{g}^{(t)} - \bar{\mathbf{g}}^{(t)} - \nabla F(\mathbf{x}^{(t)}) + \nabla f(\mathbf{x}^{(t)}) \right\|^2$$

$$\overset{(d)}{\leq} \frac{2\gamma^2}{1-\beta} \sum_{t=m\tau}^{k-1} \sum_{j=t+1}^{k-1} \beta^{j-1-t} \left\| \prod_{i=t}^{k-1} \widehat{W}^{(i)} \right\|_2^2 \mathbb{E} \left\| \mathbf{g}^{(t)} - \nabla F(\mathbf{x}^{(t)}) \right\|^2$$

$$\overset{(e)}{\leq} \frac{2\gamma^2}{1-\beta} \sum_{t=m\tau}^{k-1} \sum_{j=t+1}^{k-1} \beta^{j-1-t} \left\| \prod_{i=t}^{k-1} \widehat{W}^{(i)} \right\|_2^2 \sigma^2$$

$$\overset{(f)}{\leq} \frac{4\tau\gamma^2 \rho_{\max}^2}{(1-\beta)^2} \sigma^2 \quad (90)$$

where the step (a) is thanks to the independent properties of gradient noise; in the step (b), we apply the Jensen's inequality and loosen the sum of weights to $1/(1-\beta)$; step (c) utilized the submultiplicative property of norm; by noting that

$$\bar{\mathbf{g}}^{(t)} - \nabla f(\mathbf{x}^{(t)}) = \frac{1}{n} \mathbb{1}_n \mathbb{1}_n^T \left( \mathbf{g}^{(t)} - \nabla F(\mathbf{x}^{(t)}) \right) \quad (91)$$

step (d) applies the inequality $\|x - \bar{x}\|^2 \leq \|x\|^2$; step (e) is because of Assumption A.2; step (f) define that

$$\rho_{\max}^2 \triangleq \max_{k,t} \left\| \prod_{i=t}^{k-1} \widehat{W}^{(i)} \right\|_2^2 \quad \forall k \geq \tau, t \in [m\tau, k-1] \quad (92)$$

It is easy to that for any $i$:

$$\left\| \widehat{W}^{(i)} \right\|_2^2 = \lambda_{\max} \left( (W^{(i)})^T W^{(i)} - \frac{1}{n} \mathbb{1}_n \mathbb{1}_n^T \right) \leq 1 \quad (93)$$

where the inequality is thanks to the property of doubly stochastic matrix. (Noting $(W^{(i)})^T W^{(i)}$ is just a symmetric doubly stochastic matrix). So using the sub-multiplicity property of matrix norm, $\rho_{\max}^2$ also equals to the following definition:

$$\rho_{\max}^2 := \max_{i \in [0, \tau-1]} \left\| \widehat{W}^{(i)} \right\|_2^2 \tag{94}$$

We will revisit this quantity numerically later. In most of case, this $\rho_{\max}^2$ can be omitted since it equals to 1, but we keep it for the place-holder. Next, we can use the similar procedure for the first term in (89). The difference is that the first step, we use Jensen's inequality to take the summation over $t$ out of the norm since we can no longer use the independent assumption about the noise:

$$(A) \leq \frac{4\tau\gamma^2}{(1-\beta)^2} \sum_{t=m\tau}^{k-1} \sum_{j=t+1}^{k-1} \beta^{j-1-t} \left\| \prod_{i=t}^{k-1} \widehat{W}^{(i)} \right\|_2^2 \mathbb{E} \|\nabla F(\mathbf{x}^{(t)}) - \nabla f(\mathbf{x}^{(t)})\|^2$$

$$\leq \frac{4\tau\gamma^2 \rho_{\max}^2}{(1-\beta)^2} \sum_{t=m\tau}^{k-1} \mathbb{E} \|\nabla F(\mathbf{x}^{(t)}) - \nabla \mathcal{F}(\bar{\mathbf{x}}^{(t)}) + \nabla \mathcal{F}(\bar{\mathbf{x}}^{(t)}) - \nabla f(\bar{\mathbf{x}}^{(t)})$$

$$+ \nabla f(\bar{\mathbf{x}}^{(t)}) - \nabla f(\mathbf{x}^{(t)})\|^2$$

$$\overset{(a)}{\leq} \frac{8\tau\gamma^2 \rho_{\max}^2}{(1-\beta)^2} \sum_{t=m\tau}^{k-1} \left( \mathbb{E} \|\nabla F(\mathbf{x}^{(t)}) - \nabla \mathcal{F}(\bar{\mathbf{x}}^{(t)}) + \nabla f(\bar{\mathbf{x}}^{(t)}) - \nabla f(\mathbf{x}^{(t)})\|^2 \right.$$

$$\left. + \mathbb{E} \|\nabla \mathcal{F}(\bar{\mathbf{x}}^{(t)}) - \nabla f(\bar{\mathbf{x}}^{(t)})\|^2 \right)$$

$$\overset{(b)}{\leq} \frac{8\tau\gamma^2 \rho_{\max}^2}{(1-\beta)^2} \sum_{t=m\tau}^{k-1} \left( \mathbb{E} \|\nabla F(\mathbf{x}^{(t)}) - \nabla \mathcal{F}(\bar{\mathbf{x}}^{(t)})\|^2 + \mathbb{E} \|\nabla \mathcal{F}(\bar{\mathbf{x}}^{(t)}) - \nabla f(\bar{\mathbf{x}}^{(t)})\|^2 \right)$$

$$\overset{(c)}{\leq} \frac{8\tau\gamma^2 \rho_{\max}^2}{(1-\beta)^2} \sum_{t=m\tau}^{k-1} \left( L^2 \mathbb{E} \|\mathbf{x}^{(t)} - \bar{\mathbf{x}}^{(t)}\|^2 + b^2 \right)$$

$$\leq \frac{8\tau\gamma^2 \rho_{\max}^2 L^2}{(1-\beta)^2} \sum_{t=m\tau}^{k-1} \mathbb{E} \|\mathbf{x}^{(t)} - \bar{\mathbf{x}}^{(t)}\|^2 + \frac{16\tau^2\gamma^2 \rho_{\max}^2}{(1-\beta)^2} b^2 \tag{95}$$

where step (a) applied Jensen's inequality; step (b) is similar as (91) by applying the inequality $\|x - \bar{x}\|^2 \leq \|x\|^2$; step (c) utilize the $L$-smoothness assumption and the data heterogeneous assumption (Assumption A.1 and A.3);

Plugging (90) and (95) back to (89), we establish

$$\mathbb{E} \left\| \mathbf{x}^{(k)} - \bar{\mathbf{x}}^{(k)} \right\|^2 \leq \frac{8\tau\gamma^2 \rho_{\max}^2 L^2}{(1-\beta)^2} \sum_{t=m\tau}^{k-1} \mathbb{E} \|\mathbf{x}^{(t)} - \bar{\mathbf{x}}^{(t)}\|^2 + \frac{4\tau\gamma^2 \rho_{\max}^2}{(1-\beta)^2} (\sigma^2 + 4\tau b^2), \quad \forall k \geq \tau \tag{96}$$

Taking average over iteration $k$ from 0 to $T$, we have

$$\frac{1}{(T+1)} \sum_{k=0}^{T} \mathbb{E} \left\| \mathbf{x}^{(k)} - \bar{\mathbf{x}}^{(k)} \right\|^2$$

$$\leq \frac{8\tau\gamma^2 \rho_{\max}^2 L^2}{(1-\beta)^2} \frac{1}{T+1} \sum_{k=\tau}^{T} \sum_{t=m\tau}^{k-1} \mathbb{E} \|\mathbf{x}^{(t)} - \bar{\mathbf{x}}^{(t)}\|^2 + \frac{4\tau\gamma^2 \rho_{\max}^2}{(1-\beta)^2} (\sigma^2 + 4\tau b^2)$$

$$+ \frac{1}{(T+1)} \sum_{k=0}^{\tau-1} \mathbb{E} \left\| \mathbf{x}^{(k)} - \bar{\mathbf{x}}^{(k)} \right\|^2 \tag{97}$$

One key observation is that for arbitrary term $\psi_t$, there exists a non-negative sequence $\{d_k\}$ which is uniformly bounded by $2\tau$ such that

$$\sum_{k=\tau}^{T} \sum_{t=m\tau}^{k-1} \psi_t = \sum_{k=0}^{T} d_k \psi_k, \quad \forall \psi_t \tag{98}$$

It implies

$$\frac{1}{T+1}\sum_{k=0}^{T}\mathbb{E}\left\|\mathbf{x}^{(k)}-\bar{\mathbf{x}}^{(k)}\right\|^2 \leq \frac{16\tau^2\gamma^2\rho_{\max}^2 L^2}{(1-\beta)^2}\frac{1}{T}\sum_{k=0}^{T}\mathbb{E}\|\mathbf{x}^{(k)}-\bar{\mathbf{x}}^{(k)}\|^2 + \frac{4\tau\gamma^2\rho_{\max}^2}{(1-\beta)^2}(\sigma^2+4\tau b^2)$$
$$+ \frac{1}{(T+1)}\sum_{k=0}^{\tau-1}\mathbb{E}\left\|\mathbf{x}^{(k)}-\bar{\mathbf{x}}^{(k)}\right\|^2 \tag{99}$$

We can conclude that

$$\frac{1}{T+1}\sum_{k=0}^{T}\mathbb{E}\left\|\mathbf{x}^{(k)}-\bar{\mathbf{x}}^{(k)}\right\|^2 \leq \left(1-\frac{16\tau^2\gamma^2\rho_{\max}^2 L^2}{(1-\beta)^2}\right)^{-1}\frac{4\tau\gamma^2\rho_{\max}^2}{(1-\beta)^2}(\sigma^2+4\tau b^2) \tag{100}$$
$$+ \left(1-\frac{16\tau^2\gamma^2\rho_{\max}^2 L^2}{(1-\beta)^2}\right)^{-1}\frac{1}{(T+1)}\sum_{k=0}^{\tau-1}\mathbb{E}\left\|\mathbf{x}^{(k)}-\bar{\mathbf{x}}^{(k)}\right\|^2$$

where the step-size $\gamma$ has to be small enough. Supposing

$$\frac{16\tau^2\gamma^2\rho_{\max}^2 L^2}{(1-\beta)^2} \leq \frac{1}{2} \implies \gamma \leq \frac{1-\beta}{6L\tau\rho_{\max}} \tag{101}$$

it guarantees that

$$\frac{1}{T+1}\sum_{k=0}^{T}\mathbb{E}\left\|\mathbf{x}^{(k)}-\bar{\mathbf{x}}^{(k)}\right\|^2 \leq \frac{8\tau\gamma^2\rho_{\max}^2}{(1-\beta)^2}(\sigma^2+4\tau b^2) + \frac{2}{(T+1)}\sum_{k=0}^{\tau-1}\mathbb{E}\left\|\mathbf{x}^{(k)}-\bar{\mathbf{x}}^{(k)}\right\|^2 \tag{102}$$

Since $\rho_{\max} \leq 1$, (101) can be further relaxed into the condition $\gamma \leq \frac{1-\beta}{6L\tau}$. ∎

As we seen in (102), there is an extra terms of $\sum_{k=0}^{\tau-1}\mathbb{E}\left\|\mathbf{x}^{(k)}-\bar{\mathbf{x}}^{(k)}\right\|^2$ due to the initial phase. But it is easy to see the impact of this is small since it only contain the initial $\tau$-iterations results and coefficient is diminished by $T$. When the $T$ is large enough, the extra term is almost negligible. Moreover, we can use some warm-up strategy, such as allreduce, that forces all agents' iterates in the first period are the same, i.e. $\sum_{k=0}^{\tau-1}\mathbb{E}\left\|\mathbf{x}^{(k)}-\bar{\mathbf{x}}^{(k)}\right\|^2 = 0$. Under this situation, we immediately obtain the following corollary.

**Corollary 3** *Under the same assumptions as lemma 6 and using the all-reduce warm-up strategy at the first $\tau$ iterations, it holds*

$$\frac{1}{T+1}\sum_{k=0}^{T}\mathbb{E}\left\|\mathbf{x}^{(k)}-\bar{\mathbf{x}}^{(k)}\right\|^2 \leq \frac{8\tau\gamma^2\rho_{\max}^2}{(1-\beta)^2}(\sigma^2+4\tau b^2) \tag{103}$$

**Proof.** Replacing $\sum_{k=0}^{\tau-1}\mathbb{E}\left\|\mathbf{x}^{(k)}-\bar{\mathbf{x}}^{(k)}\right\|^2$ by 0 gives the conclusion immediately. ∎

Lastly, we revisit the quantity $\rho_{\max}^2$ by numerical experiment here. Looking at the (90) again, we relax our bounds by simply taking the maximum value of all $\|\prod_{i=t}^{k-1}\widehat{W}^{(i)}\|_2^2$. But this value can be much smaller than $\rho_{\max}^2$. We just validate them by the numerical experiment in Fig. 12.

### D.4 Proof of the convergence Theorem 1

Finally, we are ready to present the convergence theorem about the decentralized momentum SGD over one-peer exponential graph. Substituting the conclusion of the descent lemma 4 into the consensus lemma6, we immediately establish

$$\frac{1}{T+1}\sum_{k=1}^{T}\mathbb{E}\|\nabla f(\bar{\mathbf{x}}^{(k)})\|^2 \leq \frac{2(1-\beta)}{\gamma(T+1)}(\mathbb{E}f(\bar{\mathbf{z}}^{(0)})-f^\star) + \frac{\gamma L}{n(1-\beta)}\sigma^2 + \frac{\beta L\gamma}{n(1-\beta)^2}\sigma^2$$
$$+ \frac{8\tau\gamma^2 L^2\rho_{\max}^2}{(1-\beta)^2}(\sigma^2+4\tau b^2) + \frac{2L^2}{(T+1)}\sum_{k=0}^{\tau-1}\mathbb{E}\left\|\mathbf{x}^{(k)}-\bar{\mathbf{x}}^{(k)}\right\|^2 \tag{104}$$

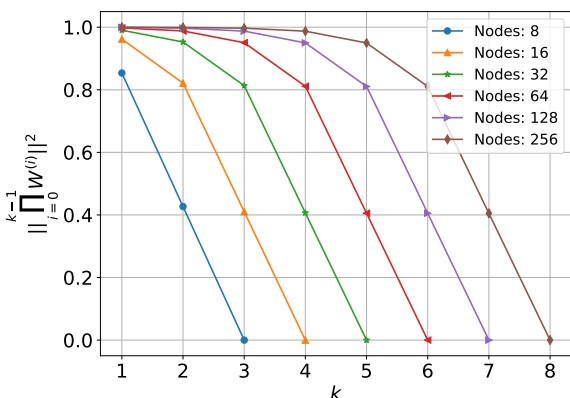

Figure 12: The value of $\|\prod_{i=0}^{k-1} \widehat{W}^{(i)}\|^2$ evolves with $k$ over different number of nodes.

where the learning rate $\gamma$ requires:

$$\gamma \leq \min\left\{\frac{1-\beta}{6L\tau}, \frac{(1-\beta)^2}{2(1+\beta)L}\right\} \tag{105}$$

Simplifying and grouping the terms, we obtain

$$\frac{1}{T+1}\sum_{k=1}^{T}\mathbb{E}\,\|\nabla f(\bar{\mathbf{x}}^{(k)})\|^2$$

$$\leq \frac{2(1-\beta)}{\gamma(T+1)}\left(\mathbb{E}\,f(\bar{\mathbf{z}}^{(0)}) + \frac{\gamma L^2}{1-\beta}\sum_{k=0}^{\tau-1}\mathbb{E}\,\left\|\mathbf{x}^{(k)} - \bar{\mathbf{x}}^{(k)}\right\|^2 - f^\star\right)$$

$$+ \frac{\gamma L}{n(1-\beta)^2}\sigma^2 + \frac{8\tau\gamma^2 L^2\rho_{\max}^2}{(1-\beta)^2}(\sigma^2 + 4\tau b^2)$$

$$= O\left(\frac{(1-\beta)}{\gamma T}\right) + O\left(\gamma\frac{\sigma^2}{n(1-\beta)^2}\right) + O\left(\frac{\sigma^2\tau\gamma^2}{(1-\beta)^2}\right) + O\left(\frac{b^2\tau^2\gamma^2}{(1-\beta)^2}\right) \tag{106}$$

If we set the learning rate as $\gamma = O\left(\frac{\sqrt{n(1-\beta)^3}}{\sqrt{T}}\right)$, we have

$$\frac{1}{T}\sum_{k=1}^{T}\mathbb{E}\,\|\nabla f(\bar{\mathbf{x}}^{(k)})\|^2 = O\left(\frac{\sigma^2}{\sqrt{(1-\beta)nT}}\right) + O\left(\frac{n(1-\beta)\sigma^2\tau}{T}\right) + O\left(\frac{n(1-\beta)b^2\tau^2}{T}\right) \tag{107}$$

Last, we derive the transient iteration complexity for the data-homogeneous and data-heterogeneous scenarios, respectively.

$$\frac{\sigma^2}{\sqrt{(1-\beta)nT}} = \frac{n(1-\beta)\sigma^2\tau}{T} \implies T = (1-\beta)^3 n^3\tau^2 \text{ (data-homogeneous)} \tag{108}$$

$$\frac{\sigma^2}{\sqrt{(1-\beta)nT}} = \frac{n(1-\beta)b^2\tau^2}{T} \implies T = (1-\beta)^3 n^3\tau^4(b^4/\sigma^4) \text{ (data-heterogeneous)} \tag{109}$$

Absorbing the constants into $\Omega(\cdot)$ notation and replacing $\tau$ by $\log_2(n)$, we establish the transient iteration complexity as stated in Theorem 1. ∎

### D.5 Comparison with other commonly-used graphs

#### D.5.1 Comparison in per-iteration communication and transient iteration (Homogeneous)

Table 7 summarizes the per-iteration communication and transient iteration complexity of DmSGD with commonly-used topologies. The details of each topology and its associated weight matrix $W$

can be referred to Sec. A.3. Table 7 assumes homogeneous data distributions across all nodes. If the logarithm term can be ignored when $n$ is large, it is observed that both static and one-peer exponential graphs can achieve $\tilde{\Omega}(1)$ per-iteration communication and $\tilde{\Omega}(n^3)$ transient iteration complexity, both of which are nearly best among all compared graphs. Table 7 is an extension of Table 1 by comparing with more topologies.

Table 7: Comparison in per-iteration communication time and transient iteration complexity between decentralized momentum SGD over various commonly-used topologes. The table assumes **homogeneous** data distributions across all nodes.

|  | **Per-iter. Comm.** | **Transient iter. complexity** |
|---|---|---|
| ring | $\Omega(2)$ | $\Omega(n^7)$ |
| star graph | $\Omega(n)$ | $\Omega(n^7)$ |
| 2D-Grid | $\Omega(4)$ | $\Omega(n^5 \log_2^2(n))$ |
| 2D-Torus | $\Omega(4)$ | $\Omega(n^5)$ |
| $\frac{1}{2}$-random graph | $\Omega(\frac{n}{2})$ | $\Omega(n^3)$ |
| bipartite random match | $\Omega(1)$ | N.A. |
| static exponential | $\Omega(\log_2(n))$ | $\Omega(n^3 \log_2^2(n))$ |
| one-peer exponential | $\Omega(1)$ | $\Omega(n^3 \log_2^2(n))$ |

#### D.5.2 Comparison in per-iteration communication and transient iteration (Heterogeneous)

Table 8 summarizes the per-iteration communication and transient iteration complexity of DmSGD with commonly-used topologies when data distributions are heterogeneous. Compared to Table 7, it is observed that the transient iteration complexity achieved by each topology in the heterogeneous scenario is typically worse than that in the heterogeneous scenario. Again, if the logarithm term can be ignored when $n$ is large, it is observed that both static and one-peer exponential graphs can achieve $\tilde{\Omega}(1)$ per-iteration communication and $\tilde{\Omega}(n^3)$ transient iteration complexity, both of which are nearly best among all compared graphs.

Table 8: Comparison in per-iteration communication time and transient iteration complexity between decentralized momentum SGD over various commonly-used topologes. The table assumes **heterogeneous** data distributions across all nodes.

|  | **Per-iter. Comm.** | **Transient iter. complexity** |
|---|---|---|
| ring | $\Omega(2)$ | $\Omega(n^{11})$ |
| star graph | $\Omega(n)$ | $\Omega(n^{11})$ |
| 2D-Grid | $\Omega(4)$ | $\Omega(n^7 \log_2^4(n))$ |
| 2D-Torus | $\Omega(4)$ | $\Omega(n^7)$ |
| $\frac{1}{2}$-random graph | $\Omega(\frac{n}{2})$ | $\Omega(n^3)$ |
| bipartite random match | $\Omega(1)$ | N.A. |
| static exponential | $\Omega(\log_2(n))$ | $\Omega(n^3 \log_2^4(n))$ |
| one-peer exponential | $\Omega(1)$ | $\Omega(n^3 \log_2^4(n))$ |

#### D.5.3 Exponential graphs endow DmSGD with smaller transient iterations: numerical validation

In Tables 7 and 8, it is observed that exponential graphs endow DmSGD with smaller transient iterations. In this subsection, we validate it with numerical experiments.

We consider a distributed logistic regression problem with each local cost function as

$$f_i(x) = \frac{1}{M} \sum_{m=1}^{M} \ln[1 + \exp(-y_{i,m} h_{i,m})^T x], \tag{110}$$

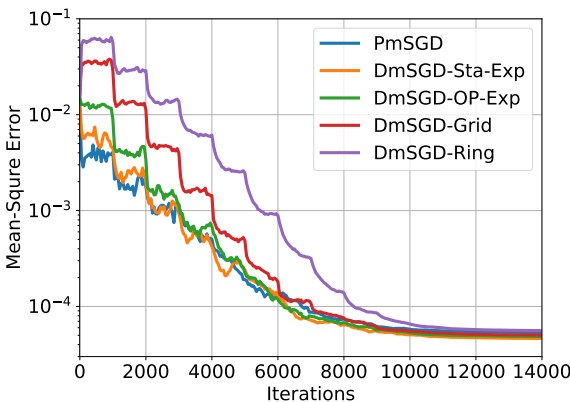

Figure 13: Convergence curves of DmSGD with various topologies. It is observed that DmSGD with exponential graphs has less transient iterations than with other graphs.

where $\{h_{i,m}, y_{i,m}\}_{m=1}^{M}$ are local data samples at agent $i$ with $h_{i,m} \in \mathbb{R}^d$ being the feature vector and $y_{i,m} \in \{+1, -1\}$ being the corresponding label. Each $h_{i,m}$ is generated from the normal distribution $\mathcal{N}(0; 10I_d)$. To generate $y_{i,m}$, we first generate an auxiliary random vector $x_i^\star \in \mathbb{R}^d$ with each entry following $\mathcal{N}(0, 1)$. Next, we generate $y_{i,m}$ from a uniform distribution $\mathcal{U}(0, 1)$. If $y_{i,m} \leq 1/[1 + \exp(-h_{i,m}^T x_i^\star)]$ then $y_{i,m}$ is set as $+1$; otherwise $y_{i,m}$ is set as $-1$. We consider a non-iid scenario in which $x_i^\star \neq x_j^\star \ \forall i, j$. Each $x_i^\star$ is normalized. We set the number of nodes as 64.

Fig. 13 illustrates the convergence curves of DmSGD with different topologies as well as parallel momentum SGD (PmSGD). The momentum parameter $\beta = 0.8$. The mean-square-error in the $y$-axis indicates $\frac{1}{n} \sum_{i=1}^{n} \mathbb{E}\|x_i^{(k)} - x^\star\|^2$. We set $d = 10$ and $M = 14000$. The step-size $\gamma$ is initialized as $0.2$ and gets decreased by half for every 1000 iterations. We repeat all simulations 20 times and illustrate the mean of all trials. It is observed that DmSGD with static exponential graph converges closely to PmSGD, and it is with the shortest transient iterations. Also, DmSGD with one-peer exponential graph is observed to have slightly longer transient iterations than with the static exponential graph. However, both exponential graphs endow DmSGD with shorter transient iterations than grid and ring.

# E  More Experiments

## E.1  Details of each topology and the weight matrix

The description of the tested topology and its associated weight matrix can be referred to Sec. A.3.1.

## E.2  DmSGD with exponential graphs when $n$ is not a power of $2$

Under the same experimental setting as in Sec. 6.2, this subsection examines the performance of exponential graphs when $n$ is not a power of 2. As shown in Table 9, one-peer exponential graph can still endow DmSGD with similar, or even better, training performance compared to its static counterpart.

Table 9: Comparison of top-1 validation accuracy(%) when using DmSGD with arbitrary numbers of nodes.

| NODES | 6(6x8 GPUs) | 9(9x8 GPUs) | 12(12x8 GPUs) | 15(15x8 GPUs) |
|---|---|---|---|---|
| STATIC EXP. | 76.21 | 75.93 | 75.73 | 76.03 |
| ONE-PEER EXP. | 76.16 | 76.17 | 75.85 | 76.19 |

## E.3  Performance with DSGD

In empirical studies, we conducted a few more experiments to validate how DSGD performs over exponential graphs in deep learning. In Table 10, we repeated the same experiment in Table 2 except for the parameter setting $\beta = 0$ i.e. eliminating the influence of momentum. It is observed that:

- The accuracy performance of all DSGD scenarios has dropped over 7% compared to the DmSGD scenarios. This highlights the critical role of the momentum in DSGD for real deep learning experiments.
- DSGD over the one-peer exponential graph achieves similar accuracy as the static exponential graph, and both topologies enable DSGD with higher accuracy than the ring topology. This is consistent with the two conclusions listed above.
- The training time of DSGD over different topologies is similar to DmSGD listed in Table 2, and we, therefore, omitted it in the following table.

Table 10: Comparison of top-1 validation accuracy(%) when using DSGD with different topologies.

| NODES | 4(4x8 GPUs) | 8(8x8 GPUs) | 16(16x8 GPUs) |
|---|---|---|---|
| RING | 68.85 | 68.62 | 68.78 |
| STATIC EXP. | 69.08 | 68.81 | 68.79 |
| ONE-PEER EXP. | 69.01 | 68.94 | 68.85 |

### E.4 Example code for implementation

For the implementation of decentralized methods, we utilize BlueFog, which is an open-source high-performance decentralized deep training framework, to facilitate the topology organization, weight matrix generation, and efficient partial averaging.

```
def neighbor_allreduce(tensor: torch.Tensor,
                       self_weight: float,
                       src_weights: Dict[int, float],
                       dst_weights: Dict[int, float]) -> torch.Tensor:
```

Listing 1: Neighbor allreduce functionality for communication.

One major functionality for decentralized communication is *neighbor_allreduce*, as listed in Listing 1, implementing the following equation.

$$x_i^{(k+1)} = w_{ii}x_i^{(k)} + \sum_{j \in \mathcal{N}_i \setminus i} w_{ij}x_j^{(k)}, \tag{111}$$

The argument *self_weight* stands for $w_{ii}$, and $w_{ij}$ for communication with the other node $j$ is achieved by either using *src_weights* in pull-mode or using *dst_weights* in push-mode.

Listing 2 gives two utility functions for one-peer exponential graphs generation. For each node, each call of function *GetOnePeerExpGraphGenerator* provides the one-peer nodes connection information for communication. Passing this connection information to *neighbor_allreduce* is achieved through updating the member variables of decentralized optimizer in *UpdateOnePeerExpGraph*.

```
# One-peer exponential graph generation
def GetOnePeerExpGraphGenerator(size, self_rank):
    tau = math.ceil(math.log2(size)) # Periodic cycle
    index = 0
    while True:
        send_rank = (self_rank + 2**index) % size
        recv_rank = (self_rank - 2**index) % size
        yield send_rank, recv_rank
        index += 1
        index = index % tau

# Graph update in each iteration
one_peer_exp_graph_gen = GetOnePeerExpGraphGenerator(bf.size(), bf.rank())
def UpdateOnePeerExpGraph(optimizer):
    dst_rank, src_rank = next(one_peer_exp_graph_gen)
    optimizer.dst_weights = {dst_rank: 1.0}
    optimizer.src_weights = {src_rank: 0.5} # Corresponds to W matrix
    optimizer.self_weight = 0.5 # Corresponds to the diagonal of W matrix
```

Listing 2: Utility functions for the generation of one-peer exponential graphs.

With that, Listing 3 shows a simplified code for model training. The overall code structure is similar as the traditional model training script, with few modifications for decentralized environment. On line 5, a decentralized optimizer wraps the original SGD optimizer. Under the hood, it registers the neighbor_allreduce communication function through the hook mechanism. On line 11, the one-peer exponential graphs get updated in each iteration. After the model forward propagation is computed locally, the backward propagation is performed on line 19. Meanwhile, it also triggers communication using *neighbor_allreduce*. In order to boost the training efficiency, the time of computation and communication are overlapped as much as possible through the multi-threading. Finally, line 21 updates the model until the communication finishes.

```python
import bluefog.torch as bf
... # Model and data preparation
# Generate decentralized optimizer
optimizer = optim.SGD(model.parameters(), lr=lr, momentum=momentum)
optimizer = DmSGDOptimizer(optimizer, model=model)
...
for epoch in range(num_epochs):
    # Training the model
    for data, target in train_loader:
        # Graph update in each iteration
        UpdateOnePeerExpGraph(optimizer)
        data, target = data.cuda(), target.cuda()
        optimizer.zero_grad()
        # Local forward propagation
        output = model(data)
        loss = F.cross_entropy(output, target_batch)
        # Local backward propagation
        # Meanwhile triggering neighbor_allreduce communication
        loss.backward()
        # Model update and wait until the communication finishes
        optimizer.step()
    # Validation
    ...
```

Listing 3: Example of how to train a model using a DmSGD optimizer under a one-peer exponential graph. The communication graph is updated in each iteration.