# OpenReview forum: "Exponential Graph is Provably Efficient for Decentralized Deep Training"
_NeurIPS.cc/2021/Conference — NeurIPS 2021 Poster_

### Official Review · Reviewer_MdPU · 2021-07-02

**Rating:** 6
**Confidence:** 3

**Summary:**

This paper studies the exponential graph used for decentralized training in theory, and shows it effectively reduces the convergence bound and speeds up the deep learning tasks. A novel analysis for the spectral gap bound on exponential graph is given, which is tighter than the one shown in previous papers. Various experiments on large-scale vision tasks are conducted, the results seem to be aligned with the theory.

**Limitations And Societal Impact:**

This paper does not reflect any potential negative societal impact.

**Main Review:**

This paper addresses an important topic in decentralized training: how do we construct a communication graph to further improve communication complexity? The main contribution is the derivation and empirical verification on the spectral gap bound of exponential graph, which is novel.

My main concern is in the baseline comparison. As we know, decentralized training utilizes the gossip algorithm, which is a well-studied topic in the literature especially in wireless network and control theory. Having said this, the main content in this paper is invariant to deep learning, but rather a discussion on the mixing time of the gossip matrix in terms of different graphs. Table 1 (and some in the appendix) in this paper lists some of the graph, but does not include many related algorithms and analysis. For example, [1][2][3][4]; also some optimality analysis is given in [5]. The authors should consider having a detailed discussion on how exponential graph compares to these algorithms and why it should be preferred over them.

In my experience, usually when we scale up the experiments to hundreds of GPUs, the speedup will become sublinear in terms of number of GPUs since communication gradually becomes the bottleneck. However, in Table 2 we observe nearly perfect linear speed up. This seems interesting and I wonder if the authors have any insights on this.

Some minor comments:

(1) In the complexity expression, constants should be written as $\Omega(1)$ in Table 2.

(2) In the experiments, std over the accuracy should be provided.


== Reference

[1] CRITICAL RANDOM GRAPHS: DIAMETER AND MIXING TIME
https://arxiv.org/pdf/math/0701316.pdf

[2] The mixing time of the giant component of a random graph
https://arxiv.org/pdf/math/0610459.pdf

[3] The best mixing time for random walks on trees
https://arxiv.org/pdf/1410.5112.pdf

[4] Mixing Times for Random Walks on Geometric Random Graphs
https://archive.siam.org/meetings/analco05/papers/07SBoyd2.pdf

[5] Fastest Mixing Markov Chain on a Graph
https://web.stanford.edu/~boyd/papers/pdf/fmmc.pdf

**Time Spent Reviewing:**

4

---

> ### Author Response · Authors · 2021-08-10
> **Thanks for your review and references. We have done our best to address your concerns (Part 2).**
>
> 3. [Linear speedup] Thank you for bringing this interesting discussion. Our experience is as follows, and, hopefully, it can provide some insights.
>     * To achieve the linear speedup, the per-iteration communication cost should be (nearly) **independent** of the topology size. This will guarantee a **constant** communication-to-computation ratio no matter how many GPUs we use in deep training. This is one major reason why DmSGD over ring, grid, static exp., and one-peer exp. can achieve a nice linear speedup. In contrast, for the random graph $G(n,p=1/2)$ we experimented in the 4-th row of Table 2,  it has a sublinear speedup in training time. It is because the generated realization of $G(n,p=1/2)$ is rather dense (with the maximum degree on the order of $O(n)$), and the per-iteration communication cost is proportional to topology size $n$. The increasing communication-to-computation ratio with topology size causes communication bottleneck and hence the sublinear speedup.
>     * **Overlap** the communication and computation as much as possible. Communication and computation can be conducted **in parallel**. If the communication time can be totally covered by the computation time, we can get a perfect linear speedup. Imagine a simple ideal scenario in the one-peer exponential graph where the communication-to-computation ratio is 80%. In this case, we can hide the entire communication time behind the computation time, and hence achieve a perfect linear speedup. However, if we add an extra edge to the graph, then there will be one node with 2 neighbors instead. Now the communication-to-computation ratio is roughly 160% (delayed by this slowest node). In this case, the computation cannot cover the entire communication, and there will be additional communication overhead (which is roughly 60% of the computation time) that slows down the training process.
>     * Finally, **large batch size** is recommended for each computing node, as long as the model generalization performance is not degraded. With more data (i.e., the large batch size) processed in each iteration, the communication-to-computation ratio is further reduced, leading to easier overlap between communication and computation and resulting in linear speed up with less communication overhead.
>
> Minor comments:
>
> 1. Agree, we will change it to $\Omega(1)$.
> 2. We run the last two rows in Table 2 for 3 times and achieve the mean and std, see the table below. We cannot re-conduct all the experiments reported in the paper due to the limited time. But we will add std to all experiments in Table 2 in the revised paper. From the table, it can be observed that accuracy performance of the exponential graphs are quite stable with small standard deviation.
> |  topology\nodes   | 4(4x8 GPUs)  | 8(8x8 GPUs) | 16(16x8 GPUs) | 32(32x8 GPUs) |
> |  ----  | ----  | ----  | ----  | ----  |
> | Static Exp.  | 76.21±0.028  | 76.32±0.037  | 76.30±​​0.007 | 76.28±0.020  |
> | One-peer Exp.  | 76.28±0.063 | 76.47±0.035 | 76.42±0.037 | 76.30±0.062 |
> ||||||

---

> ### Author Response · Authors · 2021-08-10
> **Thanks for your review and references. We have done our best to address your concerns (Part 1).**
>
> Many thanks for bringing these valuable literature to our attention. We will definitely cite them and carefully discuss them in the revision. We have attempted to compare these topologies with exponential graphs studied in our paper as best as we can. We will be glad to clarify any further comments.
>
> **Major comments:**
>
> 1. [Gossip algorithm in deep learning] Before we compare topologies in references [1-5] with exponential graphs, we would like to summarize the difference between decentralized methods in deep learning and in wireless network and control theory, which will shed light on later comparison.
>     - (Topology size) The GPUs utilized in deep learning are typically very expensive. A topology with tens or hundreds of GPUs is already regarded as a large network. This is different from wireless networks which may consist of thousands of (relatively cheap) sensors or mobile agents. The properties that are very likely to hold for large networks with thousands of nodes (e.g. the connectivity of the random graphs in [1-4] with such a large size) may not valid for network with a small or moderate size.
>
>     - (Topology control) Decentralized deep learning is typically conducted in data-center GPU clusters. In these clusters, GPUs are connected with high-bandwidth channels (such as InfiniBand, the optical fiber), and they can be organized in any topology shape. However, the network connectivity in the wireless network is highly sensitive to the geographical location of the nodes, and the radius of their wireless signals. The topology cannot be controlled freely in the latter setting.
>
>     - (Balanced degree) Since the topology is in full control for deep learning, the topology design is very important for communication efficiency. In deep learning, we prefer topologies in which all nodes have identical degrees (i.e., the number of neighbors) so that they can finish the communication almost at the same time without waiting for the slowest one. Static and one-peer exponential graphs studied in our paper are such topologies. However, for the random graph in references [1-4], there always exists the possibility to generate a realization with highly unbalanced degrees, especially when the network size is not large.
>
>     - (Non-convexity and stochasticity) Optimization problems in deep learning are non-convex, and the gradient is typically noisy. In addition, the momentum, or even adaptive momentum such as ADAM, is widely used in deep learning. These settings bring new challenges to the analysis of decentralized methods. For example, we are not aware of the study in the control community that analyzes time-varying decentralized momentum SGD (DmSGD) for non-convex problems. That’s why we provide a detailed proof for DmSGD over the time-varying one-peer exponential graph in our paper.
>
> 2. [Comparison with random graphs] References [1-4] mainly discuss the random graphs. To compare them with exponential graphs, we focus on Erdos-Renyi graph $G(n, p)$ with $p = (1+c)*\log(n)/n$ (where $c > 0$), and 2-D geometric random graph $G(n, r)$ with $r^2 = (1+c)*\log(n)/n$. Both topologies are widely used in practice. Before we show the detailed derivation and comparison, we list the per-iteration communication cost and the transient iteration complexity for these two random graphs in the following table when $n$ is sufficiently large.
> |                 | E.-R. random graph                       | Geometric Random Graphs                  | Static Exp. Graph | One-peer Exp. Graph                                             |
> |-----------------|------------------------------------------|------------------------------------------|--------------------------|------------------------------------------------------------------------|
> | Per-iter Comm.  | $\tilde{\Omega}(1)$ in expectation        | $\tilde{\Omega}(1)$ in expectation        | $\tilde{\Omega}(1)$       | ${\Omega}(1)$                                                           |
> | Tran. iteration | $\tilde{\Omega}(n^3)$                     | $\tilde{\Omega}(n^5)$                     | $\tilde{\Omega}(n^3)$     | $\tilde{\Omega}(n^3)$                                                   |
> | Connectivity    | Connected when $n$ is sufficiently large | Connected when $n$ is sufficiently large | Connected for any $n$    | May not be connected at any iteration; but is proved to work for DmSGD |
> | Degree balance  | Degree can be highly unbalanced          | Degree can be highly unbalanced          | Identical degrees        | Identical degrees |
> ||||||
>
>     In the table, it is observed that the E.-R. and geometric random graphs are either equivalent to, or worse than, exponential graphs in either per-iteration communication, or the transient iteration complexity. Moreover, note that the per-iteration communication cost for random graphs is calculated in *expectation*. In practice, the maximum degree in both random graphs must be greater than the expected degree for each node in the table, which will lead to an even slower per-iteration communication cost than exponential graphs. With the results listed in the above table as well as the other comparison described below, we still recommend using exponential graphs in deep learning.
>
> 	A more detailed comparison between random graphs and static graphs is as follows.
>     * (Connectivity.) Random graph realizations discussed in [1-4] will be connected with high probability only when the number of nodes $n$ is sufficiently large (see footnote 1 in [4]). In other words, it is possible that they are not connected for small or moderate $n$. As we discussed in Point 1.a, GPU clusters usually have a small or moderate number of GPUs, and hence the generated random graph for such GPU clusters may not be connected and cannot guarantee consensus between GPUs. In contrast, a static exponential graph is always connected for any $n$, and it has been shown that a one-peer exponential graph works for DmSGD (or DSGD) even if it is not connected at any iteration.
>     * (Transient iter.) The transient iteration complexity of E.R and geometric random graphs are listed in the above table, and here we show the derivation details. From Lemma 1.1 in [4], the mixing time of the random graph is proportional to $1/(1-\rho)$ (up to $\log(n)$ factors) where $\rho$ is the second largest eigenvalue in magnitude of the weight matrix $W$. Recall the transient iteration complexity of DmSGD over static topology is $\Omega(n^3/(1-\rho)^2)$ for i.i.d. data scenario. In addition, we learned from these references that the Erdos-Renyi graph has mixing time $\Omega(\log^2(n))$ [2], and the geometric random graph has mixing time $\Omega(n/\log(n))$ [4]. Combining these facts, we achieve the transient iteration complexity of the Erdos-Renyi graph and the geometric random graphs are $\tilde{\Omega}(n^3)$ and $\tilde{\Omega}(n^5)$, which are no better than the static exponential graphs, respectively.
>     * (Per-iter. comm.) The per-iteration communication complexity of E.R and geometric random graphs are listed in the above table, and here we show the derivation details.  It is easy to verify that each node in the Erdos-Renyi graph has $n * p = (1+c) \log(n)$ neighbors **in expectation**. Moreover, from Lemma 2.1 in [4] we find each node has $n * r^2 = (1+c) \log(n)$ neighbors in expectation. In other words, the per-iteration communication for both topologies is not better than the static exponential graph. Furthermore, we have to emphasize that $(1+c)*\log(n)$ per-iter. comm. for both random topologies is achieved in expectation.  In practice, the maximum degree in both random graphs must be greater than $(1+c)*\log(n)$, which leads to a slower per-iteration communication than a static exponential graph.
>     * (Optimal mixing time in [5]) Reference [5] targets to find the optimal weight matrix $W$ given a symmetric topology. However, it does not show how to design the symmetric topology. Different from [5], our draft focuses on how to design the (time-varying and directed) topology, but the weight matrix simply chooses to have uniform weights. It is possible that the combination of weight design in [5] and the topology design in our paper may lead to an even better weight matrix. We will leave it as a research direction for future work.
>     * (Real experiments) In fact, we compared how the Erdos-Renyi random graph $G(n,p)$ where p = ½ performs against exponential graphs in real deep learning experiments, see Table 2. Note that the realization of $G(n,p=1/2)$ is very dense, and it is hence much slower than the static and one-peer exponential graphs in terms of the wall-clock training time. We will conduct deep learning experiments over  Erdos-Renyi random graph $G(n,p=(1+c)*\log(n)/n)$ in the future.

---

### Official Review · Reviewer_13Gd · 2021-07-09

**Rating:** 8
**Confidence:** 4

**Summary:**

The work proposes efficient communication topologies for Decentralized momentum Stochastic Gradient Descent (DmSGD)  algorithms used for training deep learning models. In particular, they focus on exponential graphs, where each nodes is connected to O(log(n)) other nodes. They analyze the spectral gap, which shows the graph connectivity, for exponential graphs and establish a logarithmic bound on it and claim that it was reported incorrectly in the literature to be constant.  Moreover, they propose to communicate over one-peer exponential graphs, in which each node communicates to a single neighbor. They theoretically show that a sequence of O(log(n))  one-peer exponential graphs achieve the same convergence rates, despite being far more efficient in terms of communication. They further confirm the effectiveness of their proposed method by running experiments on real datasets and applications and comparing them to other topologies and methods.

**Ethics Review Area:**

["I don’t know"]

**Limitations And Societal Impact:**

The work presents strong theoretical and empirical results. Here are few minor issues, I don't believe these should be ground for rejection.

(a) It would be helpful if running times were also reported in Table 3 to understand how using these topologies speed up the running time (if that is the case at all). Also, it would be helpful to see Fig. 3 with respect to time, as well.

(b) The reported metrics are quite close to each other for different methods and topologies. However, the results for the one-peer exponential graphs is generally better.

(c) The functions f_i are defined inside Eq. (1), which is a bit confusing. It would read nicer, if the authors defined the functions f_i below Eq. (1).



**Main Review:**

The paper provides efficient communication topologies for running Decentralized  momentum Stochastic Gradient Descent (DmSGD) algorithms. In these algorithms the data is distributed over some nodes, the nodes compute gradients in parallel and the gradients are averaged based on the underlying topology. In particular, they focus on exponential graphs and a one-peer variant. The main motivation for considering such topologies is that they are sparse and lead to efficient communication. They further theoretically show that the sparsity in communication does not hurt the convergence rate. Finally, they show the effectiveness of their proposed method by running experiments on real datasets and real applications and comparing them to other topologies and methods. The paper is well-presented and well-organized and provide generic methods that are highly interesting for the machine learning community.

**Time Spent Reviewing:**

3

---

> ### Author Response · Authors · 2021-08-10
> **Thanks for your encouraging review. We have addressed all your comments.**
>
> Thanks for the very positive comments. We have attempted to address all the comments as best as we can.
>
> 1. Agree. We complement the training time (unit: hours) of all experiments in Table 3 as follows. In the table, it is clear to observe that a one-peer exponential graph can consistently accelerate 15%-20% of the wall-clock training time compared to static exponential graph.
> |  topology\model   | ResNet-50-static | one-peer | diff | MobileNet-v2-static | one-peer | diff | EfficientNet-static | one-peer | diff |
> |  ----  | ----  | ----  | ----  | ----  | ----  | ----  | ----  | ----  | ----  |
> | Parallel SGD  | 7.0  | - |  - | 5.8 |  - |  - |9.0  | - |  - |
> | Vanilla DmSGD  | 6.6  | 5.5 |  -1.1   | 5.6  |  4.6 |  -1.0  | 8.4  | 6.9 |  -1.5 |
> | DmSGD  | 6.9  | 5.7 |  -1.2 | 5.7 | 4.8 |  -0.9  | 8.7 | 7.1 |  -1.6 |
> | QG-DmSGD  | 6.6  | 5.6 |  -1.0 | 5.6 | 4.6 |  -1.0 | 8.4  | 6.9 |  -1.5 |
> |||||||||||
>
>    As for plotting `Fig. 5` with respect to time (we think the reviewer had a typo `Fig. 3` in the comment), we will add a new figure of the training loss and accuracy versus the wall-clock training time in the revision. (Unfortunately, the reply in the openreview system does not  support inserting the figures.) But it is not hard to imagine the curve with respect to the time since the duration of each epoch is almost the same. If we keep the position of the red line (static exponential graph) unchanged, the relative position of the blue line (one-peer) will proportionally shrink in x-axis with the ratio as the time difference we provided in table 2.
>
> 2. Thanks for the positive comment. That is what we observed in the experiment. Besides the accuracy, it is worth noting that the critical property we value on the one-peer exponential graph is not only it has a nice convergence property but also it has the lowest communication cost. Note at any iteration, each node in one-peer graph only needs to communicate with one neighbor, which is the minimum number of nodes it has to communicate. That is the reason why in table 2 the communication time of an exponential graph is much smaller compared with random graphs or static exponential graphs.
>
> 3. Agree. We will revise it accordingly.
>
> We will be glad to clarify any further comments or concerns. Thanks again!

---

### Official Review · Reviewer_nimy · 2021-07-18

**Rating:** 7
**Confidence:** 3

**Summary:**

This paper studies static exponential and one-peer exponential graphs for Decentralized momentum SGD.  The results show that these two exponential graphs have a better balance between per-iteration communication and iteration complexity compared to the existing communication topology. Especially, they show that one-peer exponential graphs converge as fast as static exponential graphs but require less per-iteration communication. In the end, this paper provides adequate empirical evaluations of the exponential graphs in decentralized (momentum) SGD training.

**Limitations And Societal Impact:**

This paper does not focus on the potential impact.

**Main Review:**

This paper is overall technically sound and well written. The advantage of the exponential graphs claimed in this paper are well supported by theoretical analysis. The experimental results show that the one-peer exponential graph for DmSGD is very promising. I think the framework is potentially useful for deep learning training.

Comments and questions:
1.	How would these exponential graphs work for decentralized SGD? Is momentum necessary in this decentralized setting?
2.	how does the convergence bound in Corollary 1 compare with existing bounds
3.	Line 253, ‘Comparing (11) with (1)’, should it be (10) instead of (1)?
4.	Could the author comment on why the baseline Bi-RandMatch is competitive compare to the one-peer exponential graph?


---------
After reading the authors‘ response, most of my concerns have been addressed. I would like to increase my score to 7.

**Time Spent Reviewing:**

10

---

> ### Author Response · Authors · 2021-08-10
> **Thanks for your review. We have addressed your concern.**
>
> Many thanks for the valuable comments! We have attempted to address all the questions as best as we can. We will be glad to clarify any further comments.
>
> 1. The results we derived for DmSGD over static or one-peer exponential graphs are also valid for decentralized SGD (DSGD).
>     - In theory, our results in Corollary 1 and Theorem 1 are still valid when $\beta = 0$, see the comments in Remark 6. This leads to the following conclusions: a) One-peer exponential graph enables DSGD with the same convergence rate as static exponential graph in terms of the best-known bounds. b) Exponential graphs enable DSGD to achieve $\tilde{\Omega}(1)$ per-iteration communication and $\tilde{\Omega}(n^3)$ transient iterations, both of which are nearly the best among other known topologies. The above conclusions for DSGD are similar to those for DmSGD.
>
>     - In empirical studies,  we conducted a few more experiments to validate how DSGD performs over exponential graphs in deep learning. In the following table, we repeated the same experiment in Table 2 except for the parameter setting $\beta = 0$. Due to the limited time, we only conducted experiments over the ring, static exponential graph, and the one-peer exponential graph. Experiments with other topologies will be left for future work. From the table, it is observed that:
> a) The accuracy performance of all DSGD scenarios has dropped over 7% compared to the DmSGD scenarios. This highlights the critical role of the momentum in DSGD for real deep learning experiments.
> b) DSGD over the one-peer exponential graph achieves similar accuracy as the static exponential graph, and both topologies enable DSGD with higher accuracy than the ring topology. This is consistent with the two conclusions listed above.
> c) The training time of DSGD over different topologies is similar to DmSGD listed in Table 2, and we, therefore, omitted it in the following table.
> |  topology\nodes   | 4(4x8 GPUs)  | 8(8x8 GPUs) | 16(16x8 GPUs) |
> |  ----  | ----  | ----  | ----  |
> | Ring  | 68.85 | 68.62  | 68.78 |
> | Static Exp.  | 69.08 | 68.81  | 68.79 |
> | One-peer Exp.  | 69.01 | 68.94  |  68.85 |
> |||||
>
>
> 2. References on decentralized momentum SGD (DmSGD) are quite limited. To our knowledge, major references on DmSGD are [4, 17, 29, 58] cited in our paper, and
> ```
> [R1] Singh et.al., “SQuARM-SGD: Communication-Efficient Momentum SGD for Decentralized Optimization”, arXiv 2005.07041, 2020
> ```
>     Among these references, results in [4, 17, R1] are based on a much stronger assumption that each stochastic gradient has upper bounds. In contrast, our bounds in Corollary 1 are based on much weaker assumptions that the variance of gradient noise and the data heterogeneity are upper bounded. As a result, results in [4, 17, R1] are not comparable to our bound in Corollary 1.
>
>     Next, we focus on [29] and [58], which are two very recent references which claim their bounds to be state-of-the-art. For fair comparison, we utilize the static exponential graph as the topology for [29] and [58], and assume momentum $\beta$ to be a constant independent of $n$.  Under such settings, the convergence bound and transient iteration complexity comparison between Corollary 1 and [29, 58] are listed as follows.
> | Methods | Convergence rate | Transient iter. for homogeneous data | Transient iter. for heterogeneous data |
> | ---- | ---- | ---- | ---- |
> | Corollary 1 (ours) | $O(\frac{\sigma^2}{\sqrt{nT}} + \frac{n\sigma^2 \log_2(n)}{T} + \frac{n b^2 \log^2_2(n)}{T}) $ | $\Omega(n^3\log^2_2(n))$ | $\Omega(n^3\log^4_2(n))$ |
> | Reference [29] | $O\left( \frac{\sigma}{\sqrt{n T}} + \frac{\sigma^{2/3}\log_2^{1/3}(n)}{T^{2/3}} + \frac{b^{2/3} \log_2^{2/3}(n)}{T^{2/3}}\right)$ | $\Omega(n^3\log^2_2(n))$ | $\Omega(n^3\log^4_2(n))$ |
> | Reference [58] | $O(\frac{\sigma^2}{\sqrt{nT}} + \frac{n\sigma^2 \log_2(n)}{T} + \frac{n b^2 \log^2_2(n)}{T})$ | $\Omega(n^3\log^2_2(n))$ | $\Omega(n^3\log^4_2(n))$ |
> |||||
>
>     In the table, it is observed that while the convergence rates of these works look different and might be uncomparable (while [58] shares the same rate with ours, the rate in [29] is not comparable with us), the transient iteration complexity derived in Corollary 1 is of the same order as [29] and [58]. This implies that the bound (at least the transient iteration complexities derived) in Corollary 1 is also among the state-of-the-art results.
>
> 3. Many thanks for the careful review. Yes, it should be (10) instead of (1).
>
> 4. Thanks for this interesting question. We will add more discussions on the bipartite random match (Bi-rand-match) graph in the revised draft. In Table 2, it is observed that the Bi-rand-match has roughly the same training time compared to the one-peer exponential graph. However,  Bi-rand-match cannot achieve a competitive accuracy as the one-peer exponential graph. (see, e.g., one-peer exp. achieves 76.27% with 256 GPUs while Bi-rand-match just achieves 75.91%.)
>
>     - Bi-rand-match and one-peer exp. have roughly the same communication time (and hence the total training time) because they have the same maximum degree (i.e., 1) in their topologies. The plot illustration and details of the Bi-rand-match graph are listed in Appendix A.3.1. Since each node in the Bi-rand-match graph exchanges information with exactly one neighbor at any iteration (which is the same as the one-peer exp.), its per-iteration communication cost is $\Omega(1)$, which is the same as the one-peer exponential graph.
>     - The convergence rate of DmSGD over the Bi-rand-match graph is unknown in literature to our best knowledge. We guess the performance degradation of the Bi-rand-match graph is due to its less effectiveness to aggregate information. Recall that the one-peer exponential graph has a periodic global averaging property (see Lemma 1), which can significantly accelerate its averaging effectiveness. In practical deep learning experiments, we do observe that the test accuracy achieved by the Bi-rand-match graph is consistently slightly worse than the one-peer exponential graph. We will leave the examination of the convergence rate of DmSGD over Bi-rand-match as future work.

---

### Decision · Program_Chairs · 2021-09-27

**Decision:**

Accept (Poster)

**Comment:**

This paper deals with fully decentralized learning with SGD and shows that network topologies in the form of exponential graphs provide fast convergence and accurate models.

The reviews were quite positive and the author response helped to consolidate the scores. Overall, the theoretical and empirical results were found to be strong and relevant for practitioners in a increasingly popular topic. Therefore, the paper is accepted.